

# Mito-fission gene prognostic model for colorectal cancer

Chao Liu[1,2], Sheng Xu[1], Yuanyuan Liu[3], Zhixing Lu[1] and Jianrong Yang[4]

[1] Departments of Gastrointestinal, Hernia and Enterofistula Surgery, The People's Hospital of Guangxi Zhuang Autonomous Region, Nannning, Guangxi Province, China
[2] Department of General Surgery, The First Affiliated Hospital of Jinan University, Guangzhou, Guangdong Province, China
[3] Departments of Gynecology, The People's Hospital of Guangxi Zhuang Autonomous Region, Nanning, Guangxi Province, China
[4] Department of Hepatobiliary, Pancreas and Spleen Surgery, The People's Hospital of Guangxi Zhuang Autonomous Region, Nanning, Guangxi Province, China

Corresponding author
Jianrong Yang, yjr2024yjr@163.com

## ABSTRACT

**Background.** Dysregulated cellular metabolism is one of the major causes of colorectal cancer (CRC), including mitochondrial fission. Therefore, this study focuses on the specific regulatory mechanisms of mitochondrial dysfunction on CRC, which will provide theoretical guidance for CRC in the future.
**Methods.** The Cancer Genome Atlas (TCGA)-CRC dataset, GSE103479 dataset and 40 mitochondrial fission-related genes (MFRGs) were downloaded in this study. The differentially expressed genes (DEGs) were analyzed in TCGA-CRC samples. Using MFRGs scores as traits, key module genes associated with its scores were screened by weighted gene co-expression network analysis (WGCNA). Then, differentially expressed MFRGs (DE-MFRGs) were obtained by intersecting DEGs and key module genes. Next, DE-MFRGs were subjected to univariate Cox, least absolute shrinkage and selection operator (LASSO), multivariate Cox and stepwise regression analysis to scree hub genes and to construct the risk model. The risk model was validated in GSE103479. Finally, the hub genes were comprehensively investigated through a multi-faceted approach encompassing clinical characteristic analysis, Gene Set Enrichment Analysis (GSEA), immune infiltration analysis, and drug sensitivity prediction. Subsequently, the expression levels of the identified key genes were validated utilizing quantitative real-time fluorescence PCR (qRT-PCR), reinforcing the findings and ensuring their accuracy.
**Results.** The 49 DE-MFRGs were gained by intersecting 3,310 DEGs and 1,952 key module genes. Then, *CCDC68*, *FAM151A* and *MC1R* were screened as hub genes. Also, the risk model validated in GSE103479 showed that the higher the risk score, the worse the survival of CRC patients. Furthermore, T/N/M stages were differences in risk scores between subgroups of clinical characteristics. The memory CD4+ T cell and plasma cell were more significant differences in the low-risk group samples. The 51 drugs were showed a better response in the high-risk group patients. RT-qPCR validation results showed that *CCDC68* and *FAM151A* were down-regulated in CRC, while *MC1R* was up-regulated, consistent with the validation set results. And *FAM151A* and *MC1R* showed highly significant difference between CRC and normal samples ($P < 0.0001$).
**Conclusion.** In this study, we found *CCDC68*, *FAM151A* and *MC1R* as potential hub genes in CRC, and analyzed the molecular mechanism of mitochondrial affecting CRC, which would provide theoretical reference value for CRC.

## INTRODUCTION

Colorectal cancer (CRC) is one of the most prevalent malignancies of the digestive system and ranks as the second deadliest cancer globally. It is classified into Colon adenocarcinoma (COAD) and rectum adenocarcinoma (READ) based on the tumor's site of origin. Due to their shared etiology, pathogenesis, and histological features, COAD and READ are frequently grouped together under the umbrella term CRC (*Islam et al., 2022*; *Sedlak, Yilmaz & Roper, 2023*). The treatment strategy for CRC primarily relies on surgery, supplemented by chemotherapy, radiotherapy, immunotherapy, and targeted therapies (*Ghazi et al., 2022*). While advances in colectomy, chemotherapy, and immunotherapy have significantly improved the 5-year survival rate for patients with CRC, outcomes for those with advanced CRC remain poor (*Yoshino et al., 2022*). Consequently, there is an urgent need to unravel the molecular mechanisms underlying CRC and identify novel therapeutic targets to improve patient prognosis.

Mitochondria, integral to multicellular life, generate adenosine triphosphate (ATP) through oxidative phosphorylation, providing the energy required for cellular metabolism. Due to this role, they are often referred to as the "energy factories" of the cell (*Harrington et al., 2023*). Mitochondria exhibit dynamic structural changes, adjusting their fusion and fission in response to cellular metabolic demands. They also regulate their form, size, quantity, distribution, quality control, and transport within the cell to maintain energy balance (*Quiles & Gustafsson, 2022*). Disruption of mitochondrial fission leads to alterations in cellular metabolism, proliferation, and apoptosis (*Kleele et al., 2021*). Recent studies have highlighted that increased mitochondrial fission (MF) can promote tumor growth and metastasis (*Colpman, Dasgupta & Archer, 2023*). It has been demonstrated that enhanced MF facilitates the metabolic shift from glycolysis to oxidative phosphorylation, thereby supporting tumor cell survival under energy stress, indicating MF's pivotal role in regulating tumor cell metabolism (*Plecitá-Hlavatá et al., 2008*; *Westermann, 2012*). Additionally, mitochondrial fission influences the onset and progression of pancreatic ductal adenocarcinoma, Lung adenocarcinoma, and Hepatocellular carcinoma (*Rehman et al., 2012*; *Kashatus et al., 2015*; *Zhang et al., 2020*). In CRC, mitochondria regulate tumorigenesis through mechanisms such as histone acetylation (*Ohshima et al., 2022*), and carposide II has been shown to inhibit CRC development by modulating MF and NF-κB pathways (*Chen et al., 2019*). Despite these findings, the precise mechanism by which mitochondrial fission influences CRC remains unclear. Therefore, this study aims to further explore the underlying mechanisms of mitochondrial fission in CRC.

Utilizing public databases, this study systematically establishes the relationship between mitochondrial fission and CRC by constructing a risk model, investigates its biological functions, and evaluates the prognostic significance of key genes, providing a theoretical foundation for CRC treatment.
## MATERIALS & METHODS

### Data extraction

RNA-seq, clinical, and survival data were retrieved from The Cancer Genome Atlas (TCGA) database, encompassing 606 CRC tissue samples (including survival data) and 51 normal tissue samples, hereafter referred to as TCGA-CRC. Additionally, the GSE103479 dataset was extracted from the Gene Expression Omnibus (GEO) database, which included 155 CRC tissue samples with associated survival data. Two gene sets, GOBP MITOCHONDRIAL FISSION and GOBP POSITIVE REGULATION OF MITOCHONDRIAL FISSION, were selected from MSigDB using "mitochondrial fission" as the keyword, resulting in the identification of 40 mitochondrial fission-related genes (MFRGs) for subsequent analysis (Table 1).

### Identification of DEGs

In TCGA-CRC samples, differentially expressed genes (DEGs) were identified using the DESeq2 package (version 1.34.0) (*Love, Huber & Anders, 2014*), with thresholds set at $P$.adj $< 0.05$ and $\log_2 FC > 1.5$. The top 20 DEGs, sorted by $\log_2 FC$, were visualized *via* volcano and heat maps.

### Analysis of MFRGs scores

MFRG scores for each sample were calculated using the GSVA package (version 1.42.0) (*Hänzelmann, Castelo & Guinney, 2013*) to assess differences between sample groups. A rank-sum test was employed to evaluate differences in MFRG scores between the two sample groups. The impact of MFRGs on the survival of patients with CRC was evaluated by stratifying samples based on the optimal cut-off values of MFRG scores. Kaplan–Meier (KM) curves were plotted for high- and low-MFRG score groups using the survminer package (version 0.4.9) (*Li et al., 2020*). in order to understand the relationship between MFRGs scores and clinical characteristics (age (50-year cut-off), gender, ethnicity and T/N/M stage), CRC patients were divided into different clinical subgroups, and the differences in MFRGs scores between different subgroups were compared using the Wilcoxon test.

### Weighted gene co-expression network analysis

The expression matrix of all CRC samples from TCGA-CRC was analyzed using weighted gene co-expression network analysis (WGCNA) (version 1.70.3) (*Langfelder & Horvath, 2008*). Initially, outlier samples were excluded by clustering. The optimal soft threshold ($R^2 = 0.85$) was determined by evaluating the relationship between the soft threshold and the scale-free network evaluation coefficient. A phylogenetic tree of genes was then constructed based on gene similarity. Using MFRG scores as the phenotype, the correlation between the phenotype and gene modules was computed. Correlation coefficients and $P$-values were calculated, and a correlation heat map was generated. The module with the strongest correlation to the MFRG score ($R > 0.5$ and $P < 0.05$) was selected as the key module. Further gene filtering within the module was performed by setting the gene significance (GS) to 0.4 and module membership (MM) to 0.4, thereby identifying key module genes associated with the MFRG score.

**Table 1  List of 40 MFRGs.**

| Number | Symbol |
| --- | --- |
| 1 | DNM1L |
| 2 | RALBP1 |
| 3 | MTFR2 |
| 4 | LRRK2 |
| 5 | MIEF2 |
| 6 | COX10 |
| 7 | C11orf65 |
| 8 | DCN |
| 9 | PGAM5 |
| 10 | DDHD2 |
| 11 | KDR |
| 12 | MAPT |
| 13 | OPA1 |
| 14 | PRKN |
| 15 | FIS1 |
| 16 | MTFP1 |
| 17 | CYRIB |
| 18 | GDAP1 |
| 19 | MIEF1 |
| 20 | PPARG |
| 21 | MARCHF5 |
| 22 | VPS35 |
| 23 | MTFR1L |
| 24 | SPIRE1 |
| 25 | MFF |
| 26 | RALA |
| 27 | INF2 |
| 28 | PINK1 |
| 29 | TMEM135 |
| 30 | BNIP3 |
| 31 | STAT2 |
| 32 | AURKA |
| 33 | UCP2 |
| 34 | MUL1 |
| 35 | MYO19 |
| 36 | DDHD1 |
| 37 | AP3B1 |
| 38 | MCU |
| 39 | SLC25A46 |
| 40 | MTFR1 |

## Acquisition and enrichment analysis of DE-MFRGs

DE-MFRGs were identified by intersecting DEGs and the key module genes associated with MFRGs using the VennDiagram package (version 1.7.1) (*Ito & Murphy, 2013*). To explore the biological functions of the DE-MFRGs, Gene Ontology (GO) and Kyoto Encyclopedia of Genes and Genomes (KEGG) analyses were performed using the clusterProfiler package (version 4.2.2) (*Wu et al., 2021*), with a significance threshold set at $P < 0.05$.

## Risk model construction

To assess the prognostic impact of DE-MFRGs on patients with CRC in TCGA-CRC, a risk model was constructed. First, univariate Cox regression analysis was performed using the survival package (version 3.5-3) (*Deng & Thompson, 2023*) to calculate risk scores and identify significant genes ($P$-value < 0.05). In the second step, Least Absolute Shrinkage and Selection Operator (LASSO) analysis was conducted using the glmnet package (version 4.1-4) (*Friedman, Hastie & Tibshirani, 2010*) to select genes with strong correlations for constructing the optimal risk model. In the third step, the proportional hazards (PH) assumption was tested for the genes selected from the LASSO analysisd, and those with $P > 0.05$ were included in further analysis, followed by multivariate Cox regression analysis. Finally, a stepwise regression method was applied to optimize the multivariate Cox regression results by entering variables one by one and checking their significance, removing non-significant variables, thereby constructing the optimal regression equation and identifying the hub genes associated with prognosis.

## Validation of risk model

Using the expression levels of hub genes and the risk coefficients obtained from stepwise regression analysis, the risk score formula ($Riskscore = \sum_{i=1}^{n} coef(gene_i) \times expr(gene_i)$) was used to calculate the risk score for each patient, with patients with CRC categorized into high- and low-risk groups based on the median risk score. The prognostic value of the risk model was assessed by comparing survival statuses between the two risk groups, and KM survival curves were plotted using the survminer package (version 0.4.9) (*Li et al., 2020*). The model's validity was further assessed by plotting receiver operating characteristic (ROC) curves. The risk model was then validated in the GSE103479 dataset using the same methodology.

## Independent prognostic analysis and clinical characteristic correlation analysis

For univariate Cox analysis, six clinical characteristics (age, gender, race, T/N/M stages, and risk score) were included in the model. The results underwent PH assumption testing and multivariate Cox analysis to identify independent prognostic factors associated with CRC. Additionally, to evaluate the risk model's applicability for patients with CRC, a nomogram predicting 1-, 2-, and 3-year survival was constructed using the rms package (version 6.5-1) (*Sachs, 2017*). The nomogram's validity was verified by calibration curves and ROC analysis. Finally, in TCGA-CRC, differences in risk scores across the six clinical characteristics were analyzed using the rank-sum test ($P < 0.05$).

## GSEA analysis

Differential analysis between the two risk groups in TCGA-CRC was performed using DESeq2 (version 1.34.0) (*Love, Huber & Anders, 2014*), with |$\log_2$FC| calculated and ranked. GSEA was conducted using the KEGG background dataset (c2.cp.kegg.v2023.1.Hs.symbols.gmt), with a threshold set at FDR < 0.05. The top six enriched signaling pathways were visualized using the enrichplot package (version 1.18.3) (*Zhang et al., 2019*).

## Immune related analysis

To further investigate the differences in the immune microenvironment between the high- and low-risk groups in CRC samples, the relative abundance of 22 immune cell types across all TCGA-CRC samples was evaluated using the "CIBERSORT" algorithm, and immune cell proportions were compared between high- and low-risk group samples. Differences in immune cell composition between the risk groups were assessed with the "Wilcoxon" test ($P < 0.05$). The expression levels of 48 immune checkpoints were also compared between these groups, and the TIDE score, immune exclusion score, and immune dysfunction score were calculated using the TIDE database (http://tide.dfci.harvard.edu/) to evaluate immune therapy efficacy.

## Drug sensitivity analysis

The half maximal inhibitory concentration (IC50) for 138 common chemotherapeutic agents was calculated for all TCGA-CRC individuals using the "pRRophetic" algorithm (version 0.5) (*Geeleher, Cox & Huang, 2014*). Differences in IC50 values between the two risk groups were compared using the Wilcoxon test. A Spearman correlation analysis was performed to assess the relationship between the sensitivity to chemotherapeutic agents and hub gene expression ($|r| > 0.4$).

## Pan-cancer analysis

To examine the expression of hub genes across different cancer types, the Wilcoxon test was conducted using the rstatix package (version 0.7.2) with a significance threshold of $P < 0.05$. Results were visualized with the ggplot2 package (version 3.5.1). Box plots displayed differential expression of hub genes in pan-cancer samples, while violin plots illustrated the differences in hub gene expression between tumor and normal samples in TCGA-CRC.

## RNA isolation and quantitative real-time polymerase chain reaction (qRT-PCR)

Twenty tissue samples, including 10 CRC tumor and 10 normal samples, were collected, frozen, and stored at −80 °C. Each 50 mg sample was lysed with TRIzol reagent for total RNA isolation, following the manufacturer's instructions. RNA concentration was measured by NanoPhotometer N50, and its quality was assessed based on concentration, purity (A260/A280 ratio), and the amplification curve. RNA was reverse transcribed into cDNA using the SureScript First Strand cDNA Synthesis Kit (Servicebio, Wuhan, China). The qRT-PCR reactions consisted of three µL of reverse transcription product, five µL

| Table 2 | Primer sequence information for RT-qPCR. |
|---|---|
| **Primer** | **Sequence** |
| CCDC68 F | TGCCTTGTATGAGTCTACGTCC |
| CCDC68 R | AGCCCTGTTGAAGGTTTCCAC |
| FAM151A F | CTGAATGTGGAGTGGCTGGT |
| FAM151A R | TTCTGTGTCTGGGAGGGTCA |
| MC1R F | GTGTCGAAATGTCCTGGGGA |
| MC1R R | GACACCTCCTGGCATCTACC |
| Internal reference-GAPDH F | CGAAGGTGGAGTCAACGGATTT |
| Internal reference-GAPDH R | ATGGGTGGAATCATATTGGAAC |

of 2xUniversal Blue SYBR Green qPCR Master Mix, and one µL each of forward and reverse primers. The amplification protocol included an initial denaturation step at 95 °C for 1 min, followed by 40 cycles of denaturation at 95 °C for 20 s, annealing at 55 °C for 20 s, and extension at 72 °C for 30 s. Three technical replicates were performed for each sample. Primer sequences are provided in Table 2. GAPDH served as the internal control, and relative gene expression was quantified using the $2^{-\Delta\Delta CT}$ method. GraphPad Prism 5 was used for graph creation and statistical analysis, with *p*-values calculated for each comparison.

## Ethics approval and consent to participate

This study was conducted in accordance with the Declaration of Helsinki and approved by the Ethics Committee of The People's Hospital of Guangxi Zhuang Autonomous Region (2024.03.01, KY-ZC-2024-030).

## Consent to participate

Written informed consent was obtained from all participants.

## Statistical analysis

All statistical analyses were performed using R software. Differential expression analysis was conducted using the DESeq2 package, and survival analysis was carried out using the survminer package to plot KM survival curves. WGCNA was performed using the WGCNA package, and functional enrichment analysis was conducted with the clusterProfiler package. Cox regression analysis was performed using the R survival package, while LASSO regression analysis was carried out with the glmnet package. Additionally, ROC curves were plotted using the survivalROC package, and nomograms were constructed using the rms package. Immune infiltration cell analysis was conducted using the CIBERSORT algorithm, and the IC50 values for common chemotherapy and molecular targeted drugs were calculated using the pRRophetic package. Correlation analysis was performed using Spearman's correlation. For pairwise comparisons, the Wilcoxon test was applied, and $P < 0.05$ was considered statistically significant.

## RESULTS

### Identification of DEGs and characterization of the correlation between MFRGs and CRC

In TCGA-CRC samples, a total of 3,310 DEGs were identified, comprising 1,669 upregulated and 1,641 downregulated genes (Figs. 1A–1B). To investigate the relationship between MFRGs and CRC, differential analysis revealed that the MFRG score was significantly lower in CRC samples compared to normal samples (Fig. 1C). KM survival analysis showed that groups with high MFRG scores exhibited improved survival probabilities, suggesting that MFRGs may influence the survival of patients with CRC (Fig. 1D). Further analysis of clinical characteristics demonstrated significant differences in MFRG scores across various factors, including race, M stage (M0 *vs* M1), and N stage (N0 *vs* N1, N0 *vs* N2) (Fig. 1E).

### Acquisition of 49 DE-MFRGs

Sample clustering and phenotypic trait heatmaps showed no outlier samples (Fig. 2A). A soft-threshold of 7, with an $R^2$ value of 0.85, was selected to construct a scale-free network (Fig. 2B). Following average hierarchical clustering and dynamic tree clipping, 16 modules were identified (Fig. 2C). The correlation heatmap revealed that the MEgreen (Cor = 0.67), MEturquoise (Cor = 0.54), MEred (Cor = −0.52), and MEsalmon (Cor = −0.52) modules were significantly associated with CRC ($P < 0.05$) (Fig. 2D). A total of 917 genes, 3,143 genes, 907 genes, and 302 genes were derived from these modules, resulting in 1,952 hub module genes (GS = 0.4, MM = 0.4) (Figs. 2E–2H). By intersecting the 3,310 DEGs with the 1,952 hub genes, 49 DE-MFRGs were identified (Fig. 2I).

### Biological functions and signaling pathways involved in 49 DE-MFRGs

Functional enrichment analysis of the 49 DE-MFRGs revealed 15 significant terms in GO analysis, including caspase binding, helicase activity, kinetochore formation, and cell division site organization (Figs. 3A–3C). KEGG pathway analysis identified three key pathways, such as pentose and glucuronate interconversion and nucleotide sugar biosynthesis ($P < 0.05$) (Fig. 3D).

### Identification of hub gene and risk model construction

The prognostic value of the 49 DE-MFRGs was assessed by constructing a risk model. Initially, univariate Cox analysis was performed on the 49 DE-MFRGs in 606 CRC samples, yielding 10 genes associated with survival ($P < 0.05$). Of these, HIG1 hypoxia inducible domain family member 1A (*HIGD1A*), diaphanous related formin 3 (*DIAPH3*), coiled-coil domain containing 68 (*CCDC68*) and family with sequence similarity 151 member A (*FAM151A*) were identified as protective factors, while keratin associated protein 5-1 (*KRTAP5.1*), heat shock transcription factor 4 (*HSF4*), melanocortin 1 receptor (*MC1R*), Zinc finger protein 692 (*ZNF692*), Lck interacting transmembrane adaptor 1 (*LIME1*) and tweety family member 3 (*TTYH3*) were risk factors (Table 3 and Fig. 4A). LASSO analysis, using the minimum lambda value, selected seven genes (*HIGD1A*, *DIAPH3*, *CCDC68*, *FAM151A*, *HSF4*, *MC1R* and *TTYH3*) to construct the optimal risk model (Figs. 4B–4C).

Peer J

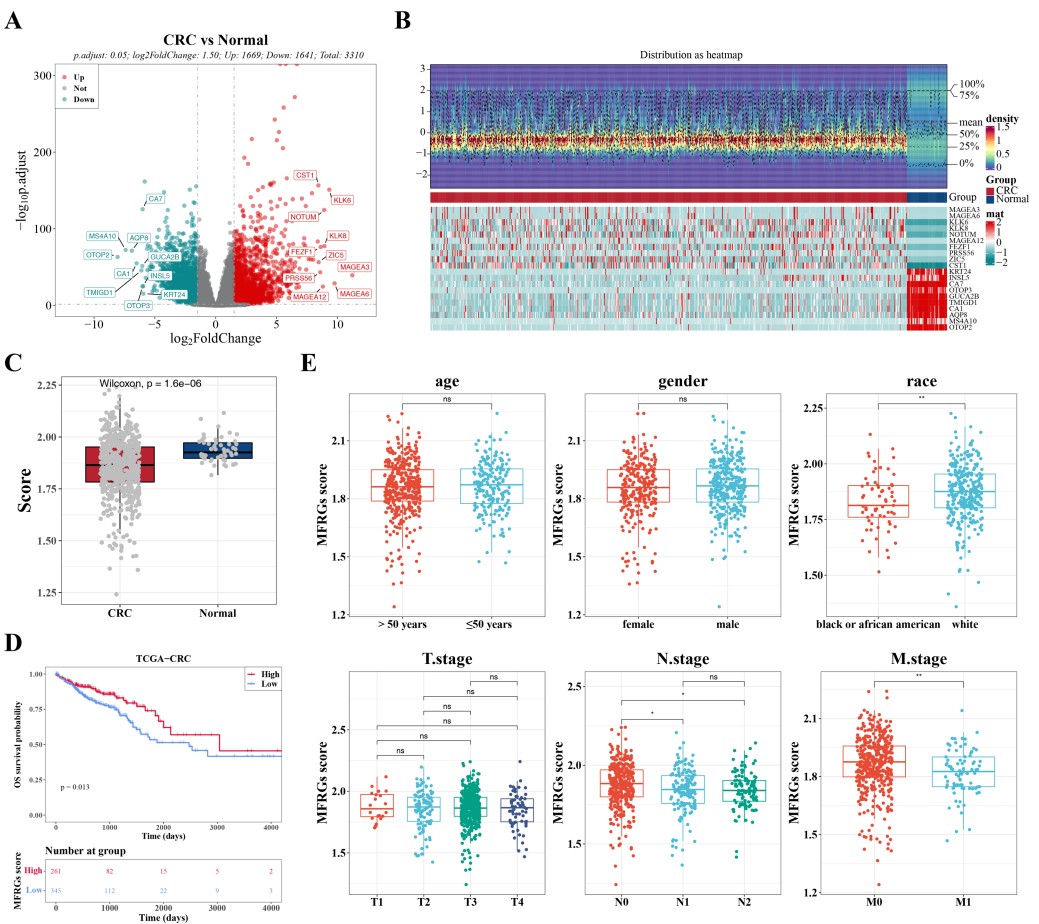

**Figure 1  The expression of DEGs between colorectal cancer group and normal group in TCGACRC samples.** (A) Volcanic map of differentially expressed genes distribution between CRC and Normal groups: red dots are up-regulated genes, green dots are down-regulated genes, and gray dots are undifferentiated genes. (B) Heat map of differentially expressed genes between the CRC group and Normal group: the upper part is the heat map of expression quantity density of the Top20 differentially expressed genes in the sample, showing the lines of five quantiles and average values; the next part is the expression heat map of the top 20 differential genes in the sample. (C) MFRGs score difference among different samples in training set. (D) KM survival curve of high and low MFRGs rating groups. (E) Differences in MFRGs scores among different clinical subgroups.

The PH assumption test for these seven genes showed *P*-values greater than 0.05, and a forest plot from multivariate Cox analysis was generated (Fig. 4D). Finally, stepwise regression analysis identified three hub genes for the construction of a prognostic risk model: *CCDC68* (HR = 0.74, 95% CI [0.56–097], *P* = 0.03) and *FAM151A* (HR = 0.38 95% CI [0.20–0.71], *P* = 0.002) were protective factors, and *MC1R* (HR = 2.22, 95% CI [1.56–3.17], *P* < 0.001) was a risk factor (Table 4 and Fig. 4E).

## Risk model had good predictability for CRC patients

In TCGA-CRC, the high- and low-risk groups divided based on the median of the risk score were calculated, with results showing a significant increase in the number of deaths as the

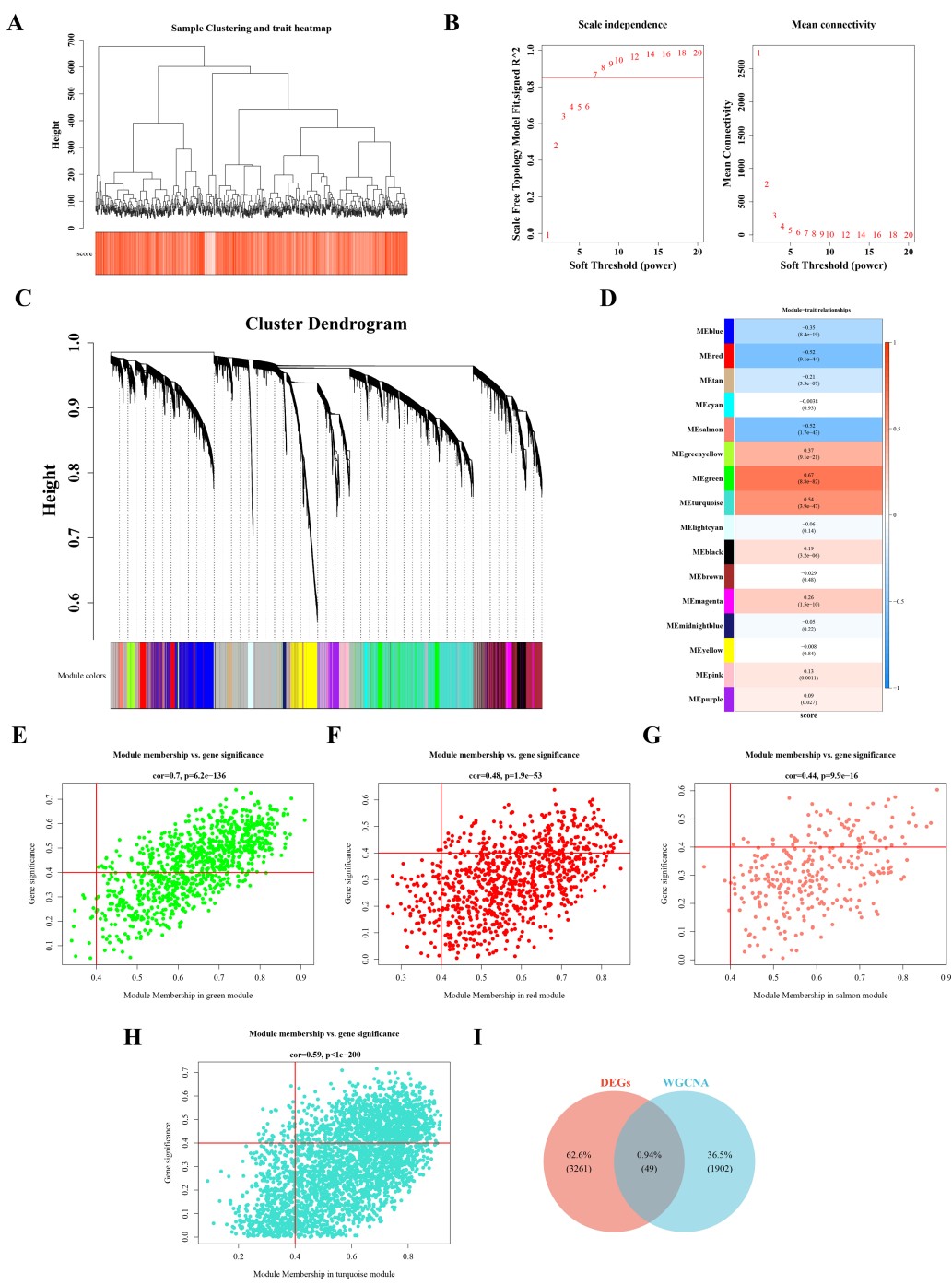

**Figure 2 Acquisition of 49 DE-MFRGs.** (A) Sample level clustering after the introduction of sample traits. (B) Soft threshold selection: the determination of the optimal soft threshold mainly refers to the figure on the left, that is, the scale-free fit index ($Y$-axis) under different soft thresholds ($X$-axis), where the red line represents the value of the selected scale-free fit index. From the figure on the left, the value when the scale-free fit index is 0.85 is the minimum soft threshold. 

risk scores rose (Figs. 5A–5B). KM analysis revealed that patients in the high-risk group had a lower survival rate (Fig. 5C). ROC analysis showed area under the curve (AUC) values of 0.631, 0.626, and 0.663 at 1-, 2-, and 3-year intervals, respectively (Fig. 5D), indicating that the risk model demonstrated reasonable predictive accuracy. The risk model was then validated in the GSE103479 dataset, with results consistent with TCGA-CRC, confirming that high-risk patients with CRC had worse survival and prognosis (Figs. 5E–5H).

### RiskScore, age, and N/M stages were independent prognostic factors for CRC

Further investigation of the correlation between clinicopathologic characteristics and the risk model in TCGA-CRC samples revealed that the risk score, along with age and T/N/M stages, showed significant association with the risk model in univariate Cox analysis ($P < 0.05$) (Fig. 6A), all of which satisfied the PH assumption ($P > 0.05$). Multivariate Cox analysis demonstrated that, except for T stage, the remaining clinicopathologic characteristics were significantly correlated with the risk model ($P < 0.05$) (Fig. 6B). A nomogram was developed to visualize the clinical characteristics, showing that higher scores correlated with increased mortality. The validity of the nomogram was confirmed by calibration curves, with an AUC exceeding 0.6, affirming its strong predictive ability for the survival of patients with CRC (Figs. 6C–6E). Further analysis of the risk scores and clinicopathologic features revealed significant differences in risk scores across T/N/M stages (Fig. 6F).

### Enrichment pathways of high- and low-risk groups and their effects in the immune micro environment

GSEA identified the top six signaling pathways significantly associated with the risk model, including cardiac muscle contraction, ascorbate and aldarate metabolism, pentose and glucuronate interconversion, retinol metabolism, extracellular matrix (ECM)-receptor interaction, and dilated cardiomyopathy (Fig. 7A).

Differences in the immune microenvironment between the high- and low-risk groups in CRC were also explored. The proportional distribution of 22 immune cell types is shown in Fig. 7B. Notably, except for M0 macrophages, resting dendritic cells, memory CD4+ T cells, activated memory CD4+ T cells, and plasma cells, the low-risk group exhibited more significant differences in immune cell proportions (Fig. 7C), indicating that the immune

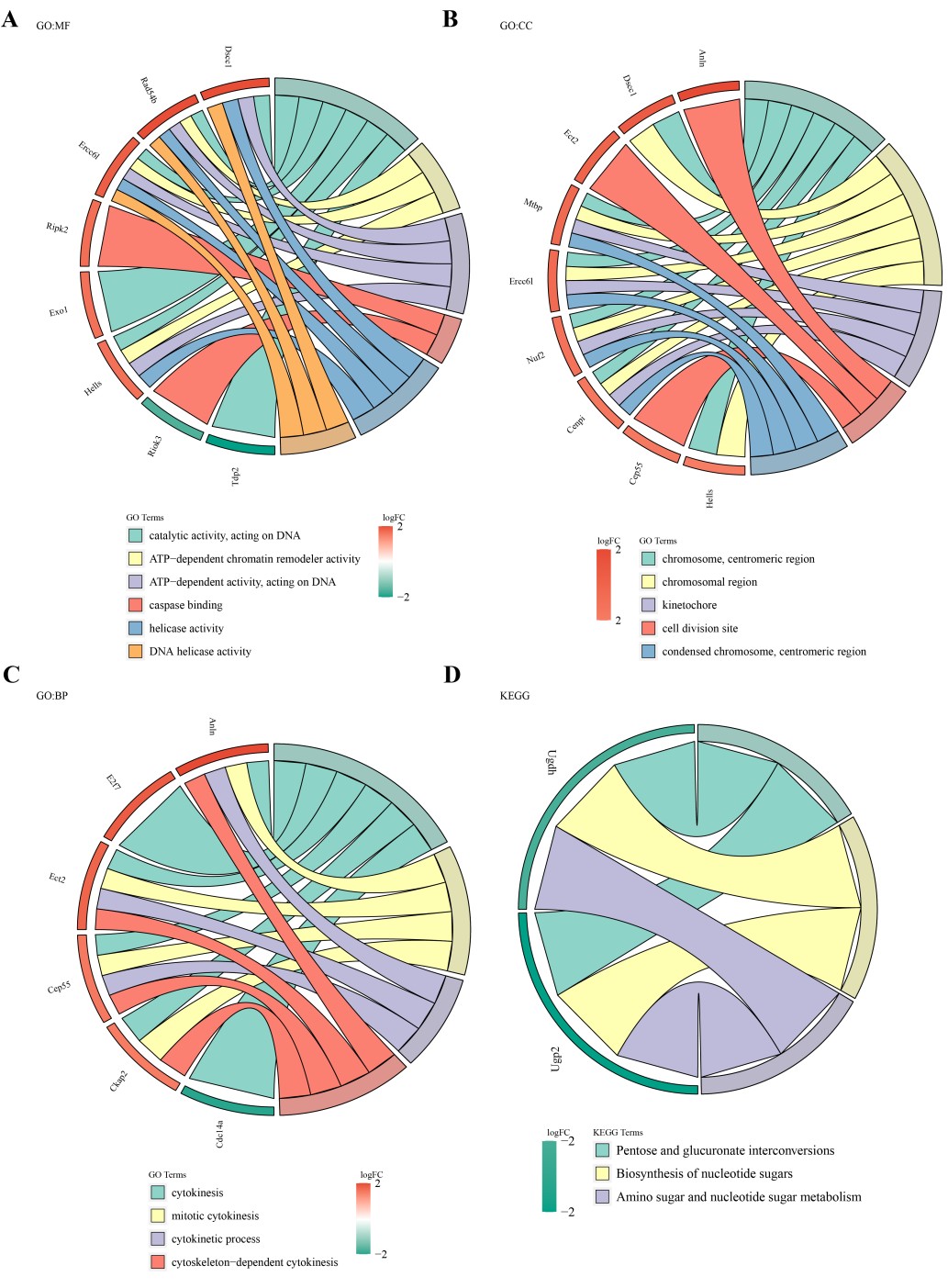

**Figure 3  Biological functions and signaling pathways involved in 49 DE-MFRGs.** (A–D) GO and KEGG enrichment results of differential MFRGS-related genes: color bands on the left represent logFC of genes, and different bands on the right represent different pathways.

**Table 3  List of functional information for ten genes from univariate Cox analysis.**

| | Full name | Function | Correlation with cancer | Correlation with mitochondrial dysregulation |
|---|---|---|---|---|
| HIGD1A | HIG1 Hypoxia Inducible Domain Family Member 1A | Regulating polyamine metabolism, inhibited tumor growth and metastasis | colon cancer (PMID: 37479180) | Mitochondrial inner membrane protein, regulating metabolic homeostasis (PMID: 37492734) |
| DIAPH3 | Diaphanous Related Formin 3 | Involved in actin remodeling and regulate cell movement and adhesion, promote migration and invasion of CRC cells | colorectal cancer (PMID: 39843730) | NA |
| CCDC68 | Coiled-Coil Domain Containing 68 | Involved in microtubule anchoring at centrosome and protein localization | colorectal cancer (PMID: 35557589) | NA |
| FAM151A | Family With Sequence Similarity 151 Member A | Highly expressed in the intestine, kidney and spleen (PMID: 31949211) | NA | NA |
| KRTAP5.1 | Keratin Associated Protein 5-1 | Essential for the formation of a rigid and resistant hair shaft through their extensive disulfide bond cross-linking with abundant cysteine residues of hair keratins (PMID: 37014946) | NA | NA |
| HSF4 | Heat Shock Transcription Factor 4 | Promotes the proliferation and metastasis of CRC by regulating EMT-related signalling pathways | colorectal cancer (PMID: 39881364) | NA |
| MC1R | Melanocortin 1 Receptor | Regulates T regulatory cell differentiation through metabolic reprogramming to promote colon cancer | colorectal cancer (PMID: 38917522) | Suppressing the oxidative stress, apoptosis, and mitochondrial fission through the AMPK/SIRT1/PGC-1α signaling pathway (PMID: 33391490) |
| ZNF692 | Zinc Finger Protein 692 | Promotes the progression of colorectal cancer by regulating HSF4 expression | colorectal cancer (PMID: 38435777) | NA |
| LIME1 | Lck Interacting Transmembrane Adaptor 1 | Docking protein to recruit signaling molecules, and involved in inflammatory pathways | bladder cancer (PMID: 39429662) | NA |
| TTYH3 | Tweety Family Member 3 | Plays a role in tissue formation, embryonic development, and immune response to pathogen related molecules | colorectal cancer (PMID: 38827027) | NA |

system of low-risk group patients was more active and more favorable for controlling tumor progression. Furthermore, 16 immune checkpoints showed significant differences between the high- and low-risk groups, including HHLA2, TNFRSF4, CD160, TNFSF4, and TIMP3 (Fig. 7D). These immune checkpoints may be involved in the activation or inhibition of immune cells, suggesting differences in immune treatment responses between the risk groups. In addition, the TIDE score and immune exclusion score of the high-risk group were significantly higher than those of the low-risk group (Fig. 7E). These two scores are more meaningful in evaluating the effectiveness of immunotherapy. The high-risk group patients, due to their higher scores, were associated with poorer prognosis, including

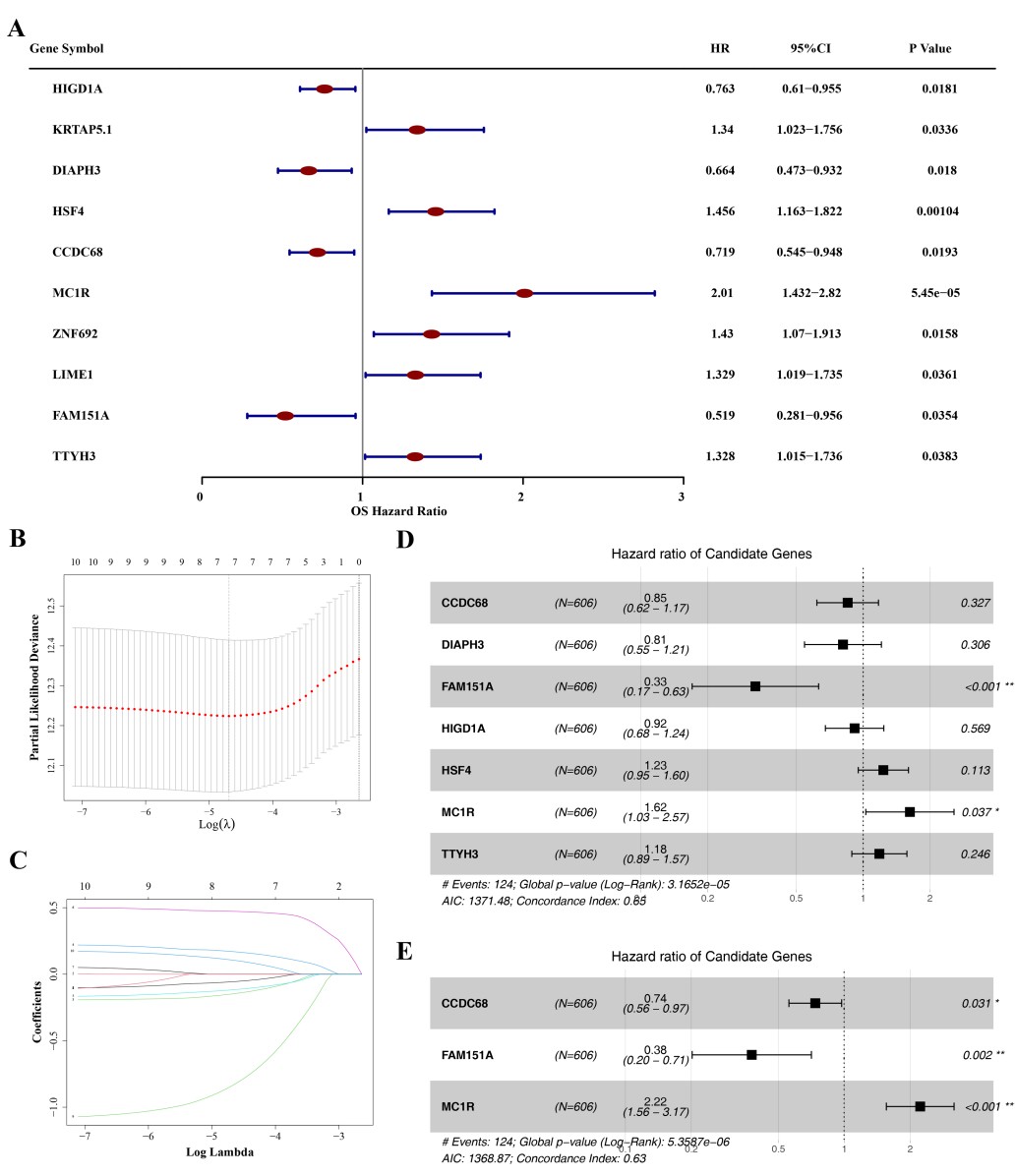

**Figure 4 Identification of *CCDC68*, *FAM151A* and *MC1R* and risk model construction.** (A) Unifactor Cox regression analysis of forest map. (B) Ten cross-validations of the adjusted parameters in LASSO analysis: the horizontal coordinate is the logarithm of the lambdas, and the vertical coordinate is the model error. The optimal lambda value is at the lowest point of the red curve, and the corresponding number of variables is 7. (C) LASSO coefficient spectrum: the horizontal coordinate is the logarithm of the lambdas, and the vertical coordinate is the variable coefficient. As the lambdas increase, the variable coefficient approaches 0. When the optimal lambda is reached, the variable whose culling coefficient is equal to 0. (D) Multivariate Cox regression analysis of forest map. (E) Stepwise regression analysis of forest map.

**Table 4 Results of stepwise regression analysis.**

| Gene | coef | exp (coef) | se (coef) | z | Pr (> |z|) |
|---|---|---|---|---|---|
| CCDC68 | −0.30553 | 0.736733 | 0.141426 | −2.16035 | 0.030746 |
| FAM151A | −0.97326 | 0.377849 | 0.31981 | −3.04325 | 0.00234 |
| MC1R | 0.797626 | 2.220264 | 0.181711 | 4.389539 | 1.14E−05 |

**Notes.**

Coef, Coefficient; exp (coef), Exponential of coefficient; se (coef), Standard error of the coefficient; $z$, $z$-value; Pr(> |$z$|), Probability of the $Z$-score being greater than its absolute value.

shorter survival and higher recurrence rates. The higher the TIDE score, the poorer the response to immune checkpoint inhibitors and the shorter the survival time.

### Fifty-one common chemotherapeutic drugs were effective in high-risk patients for CRC

The IC50 values of 71 chemotherapeutic drugs showed significant differences between the two risk group samples, with 51 drugs demonstrating better responses in the high-risk group patients (IC50 values lower in the high-risk group than in the low-risk group) (Figs. 8–9), while the remaining 20 drugs performed better in the low-risk group (Fig. 10). It was worth noting that 20 of these drugs had already been clinically applied (Table 5). This suggested a potential significant difference in chemotherapy drug sensitivity between the high-risk and low-risk groups. Spearman correlation analysis revealed that *FAM151A* was positively correlated with AG.014699, camptothecin, NVP.BEZ235, and TW.37 ($r > 0.4$), while *CCDC68* was negatively correlated with BMS.708163 and X681640 ($r < −0.4$) (Fig. 11). These findings indicate that the expression levels of certain genes may be associated with the sensitivity to specific drugs, suggesting that gene expression levels could influence tumor cell response to these drugs, or the drugs could impact gene-related functions.

### Three hub genes linked in other diseases

Pan-cancer analysis showed that *CCDC68*, *FAM151A* and *MC1R* were significantly different in thyroid cancer (THCA), renal clear cell carcinoma (RCC) and lung squamous cell carcinoma (LUSC) *etc.* (Fig. 12). In TCGA-CRC, the differential expression between CCDC68 and FAM151A were more significant in the normal group, whereas MC1R had opposite results to the above two genes.

### Validation of expression of 3 biomarkers

In TCGA-CRC, the differential expression of *CCDC68* and *FAM151A* was more significant in the normal tissue samples, whereas *MC1R* exhibited opposite expression patterns (Fig. 13A). RT-qPCR validation of hub gene expression in clinical CRC and normal tissues confirmed these findings. *CCDC68* and *FAM151A* were more highly expressed in normal samples compared to CRC samples, while *MC1R* expression was higher in CRC samples, consistent with the dataset results (Figs. 13B–13D). Notably, the expression of *FAM151A* ($P < 0.0001$) and *MC1R* ($P < 0.0001$) exhibited significant differences between clinical CRC and normal samples.

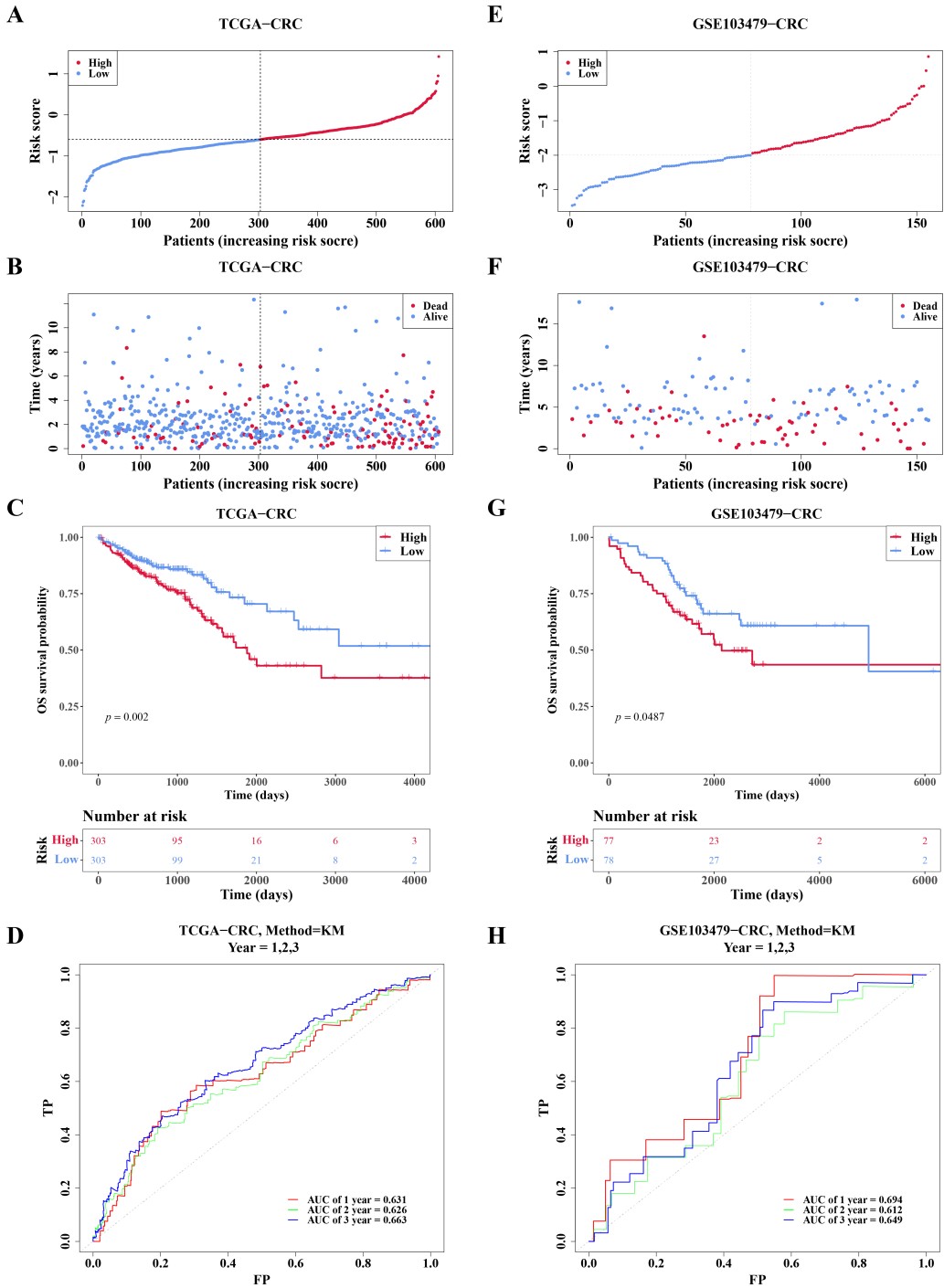

**Figure 5   Risk model had good predictability for CRC patients.** (A) Risk curves for the high-low risk group of CRC patients in the training set. (B) Scatterplot of the high-low risk grouping of CRC patients in the training set. (C) Survival curves of CRC high-low risk groups in the training set: red represents high risk group, blue represents low risk group. (D) ROC curves of CRC patients at 1, 2 and 3 years of training set. (E) Risk curves of CRC patients in high-low risk groups were validated. (F) Scatter plots of high-low risk groups of CRC patients were validated. (G) Survival curves of the high-low risk group of CRC patients in the validation set. (H) ROC curves of CRC patients at 1, 2, and 3 years were validated.

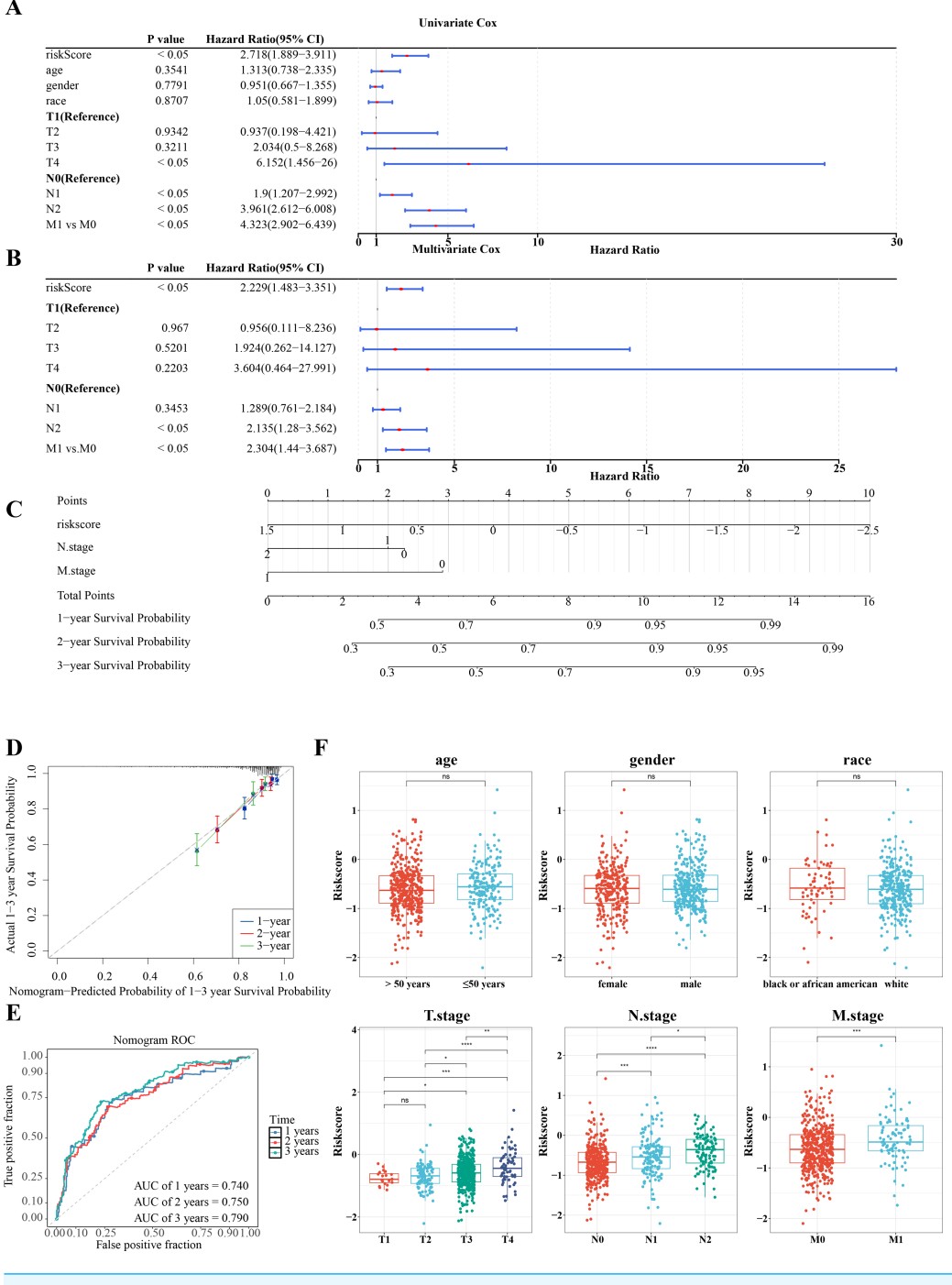

**Figure 6  RiskScore, age and N/M stages were independent prognostic factors for CRC.** (A) Univariate independent prognostic analysis of CRC forest map. (B) Multivariate independent prognostic analysis of CRC forest map. (C) The survival nomogram of CRC patients was constructed based on risk model and clinical features. (D) Calibration curve of clinical feature nomogram: the horizontal axis represents the probability of different clinical outcomes predicted by the model, (continued on next page…)

**Figure 6 (…continued)**
and the vertical axis represents the probability of actually observed clinical outcomes of patients, which is represented by the form of median plus mean, and an ideal curve with slope of 1 is drawn as a reference. The closer the actual curve is to the ideal curve, the better the calibration degree is, that is, the smaller the deviation between the predicted results of the model and the actual results, the better the model effect. (E) ROC curve of clinical features. (F) Correlation analysis between risk score and clinical features.

## DISCUSSION

The rising incidence and mortality of CRC have elevated its status as a significant public health issue (*Bray et al., 2024*). Recent studies have highlighted the pivotal role of mitochondrial fusion–fission dynamics in the initiation and progression of CRC (*Wu et al., 2024*). Dysregulated mitochondrial fission or fusion can contribute to the metabolic reprogramming of tumor cells, thereby activating oncogenic pathways that drive cell proliferation, invasion, migration, and drug resistance (*Wu et al., 2024*). This study, leveraging public databases, systematically establishes the association between mitochondrial fission and CRC by constructing a risk model, exploring its biological functions, and evaluating the prognostic efficacy of key genes, providing a theoretical foundation for CRC treatment.

GO and KEGG enrichment analyses were performed on DE-MFRGs. These analyses revealed that biological processes enriched by DE-MFRGs predominantly involve cell division and motility. Cytokinesis, a post-mitosis process, ensures the equal distribution of cell membrane, cytoskeleton, organelles, and soluble proteins to form two daughter cells. The study showed that the cortical protein *CTTN* upregulates the expression of the cytoplasmic division protein dedicator of cytokinesis 1 (*DOCK1*), and silencing *DOCK1* impairs the migration and invasion capabilities of *CTTN*. Thus, *CTTN* promotes CRC metastasis by increasing *DOCK1* expression (*Jing et al., 2016*). Furthermore, mutations in adenomatous polyposis coli (*APC*), commonly found in CRC, inhibit cell division by preventing mitotic spindle anchorage in the late cortex, thereby obstructing the initiation of the cytokinetic furrow (*Caldwell, Green & Kaplan, 2007*). Cellular components enriched by DE-MFRGs include chromosomal regions and centromere regions. Chromosomal rearrangements, such as deletions of chromosome arm 8p and amplifications of 8q, are prevalent in CRC (*Brueckner et al., 2013*). Studies on interchromosomal eight deletions in patients with CRC, using 11 microsatellite markers, revealed that eight markers are located in the centromeric region of chromosome 8p (*Chughtai et al., 1999*). Tumor suppressor genes may also be involved in a specific site of CRC, notably in a large centromere region between D11S897 and D11S925 (*Connolly et al., 1999*). These findings underscore the critical role of chromosomal and centromeric regions in CRC initiation and progression, aligning with the results of this study. Enriched molecular functions of DE-MFRGs include ATP-dependent chromatin remodeling enzyme activity, caspase binding, and DNA helicase activity. ATP-dependent chromatin remodeling enzymes, which alter nucleosome structure (*Sundaramoorthy & Owen-Hughes, 2020*), facilitate the accessibility of DNA sequences to interacting proteins, thereby enabling precise regulation of eukaryotic gene expression (*Sundaramoorthy, 2019*). Chromatin remodeling is essential for efficient DNA repair,

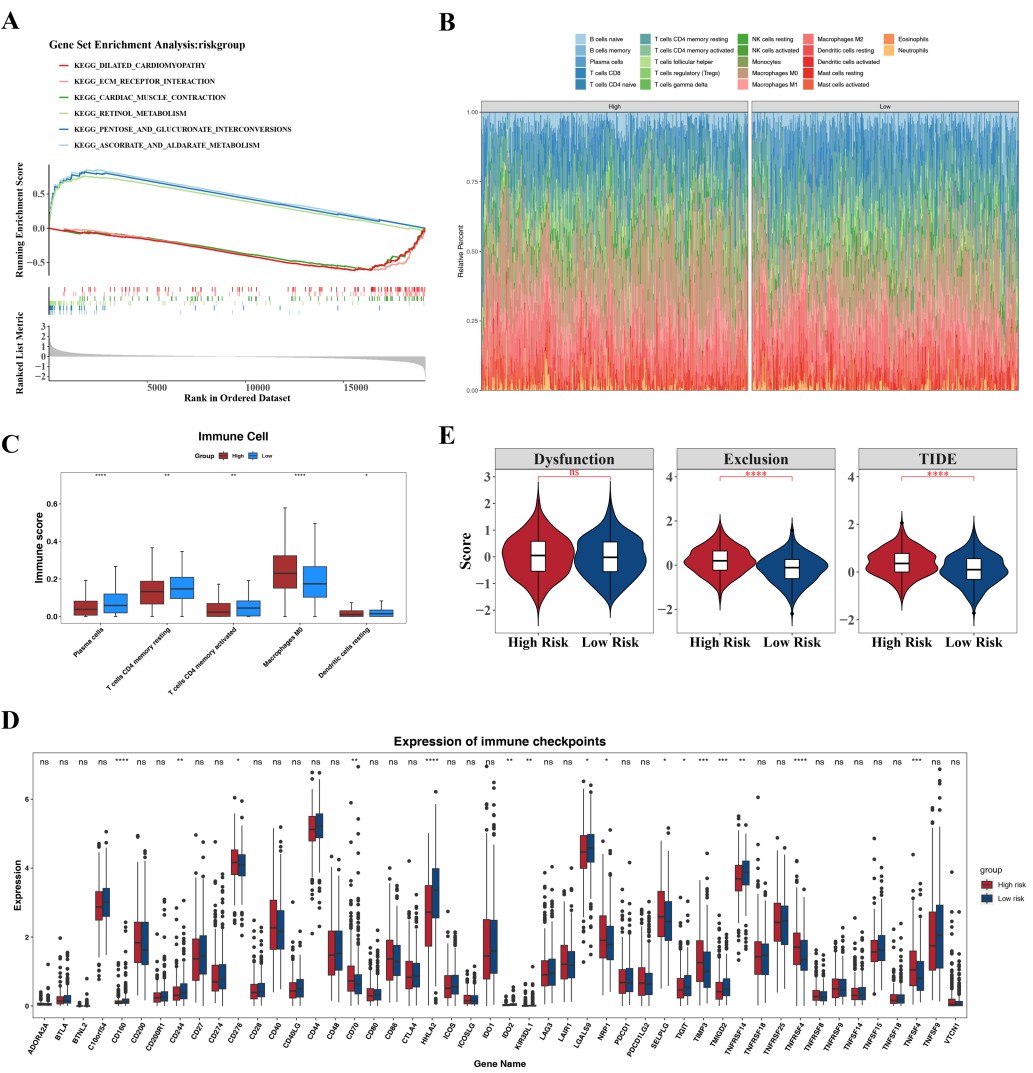

**Figure 7** **Associated pathways of three hub genes and their effects in the immune micro environment.**
(A) KEGG enrichment signaling pathway in high-low risk groups: This diagram can be divided into three parts. Part I: The top five lines are the lines of gene Enrichment Score. The vertical axis is the corresponding Running ES, and there is a peak value in the line graph, which is the Enrichemnt score of this gene set, and the genes before the peak value are the core genes under this gene set. The horizontal axis represents each gene under this gene set, corresponding to the bar code-like vertical line in the second part. Part 2: The barcode-like part, called Hits, where each vertical line corresponds to a gene under the gene set. Part 3: Sequencing of genes. (B) Proportion of immune cells in the high-low risk group. (C) Differences in immune cells between high and low risk groups. (D) Differences in 48 immune checkpoints between high and low risk groups. (E) TIDE score difference between high and low risk groups violin chart.

genome stability, and therapeutic responses. Mutations or overexpression of Helicase, lymphoid specific (*HELLS*), a member of the ATP-dependent chromatin remodeling SNF2 family, have been linked to various cancers, including CRC, hepatocellular carcinoma, and leukemia (*Peixoto et al., 2022*). Caspases, an evolutionarily conserved cysteine protease family, play pivotal roles in cell death and inflammation (*Van Opdenbosch & Lamkanfi,*

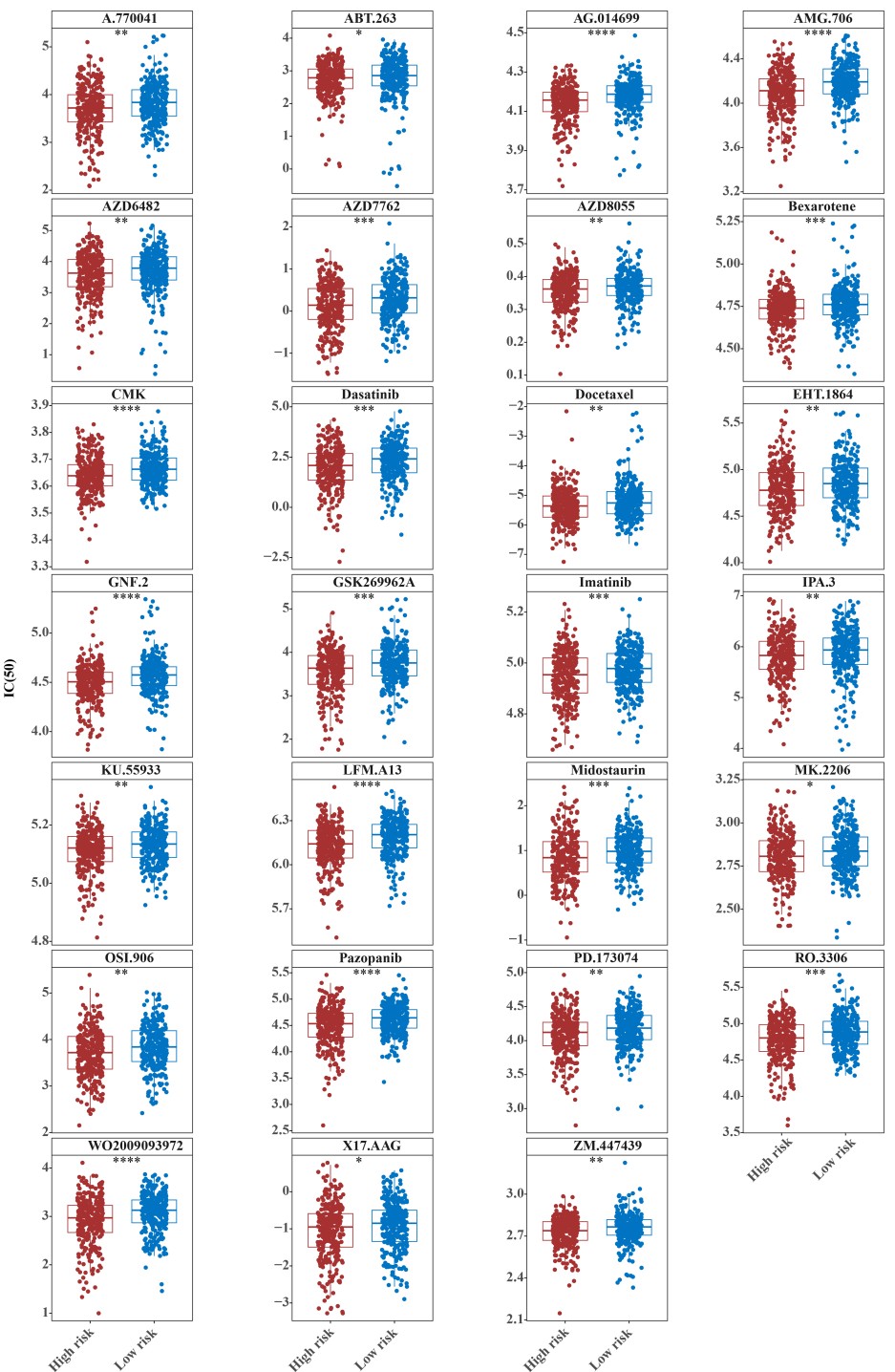

**Figure 8  Twenty-seven drugs that perform better in high-risk groups.** The horizontal coordinate is the high-low expression group; the ordinate is IC50.

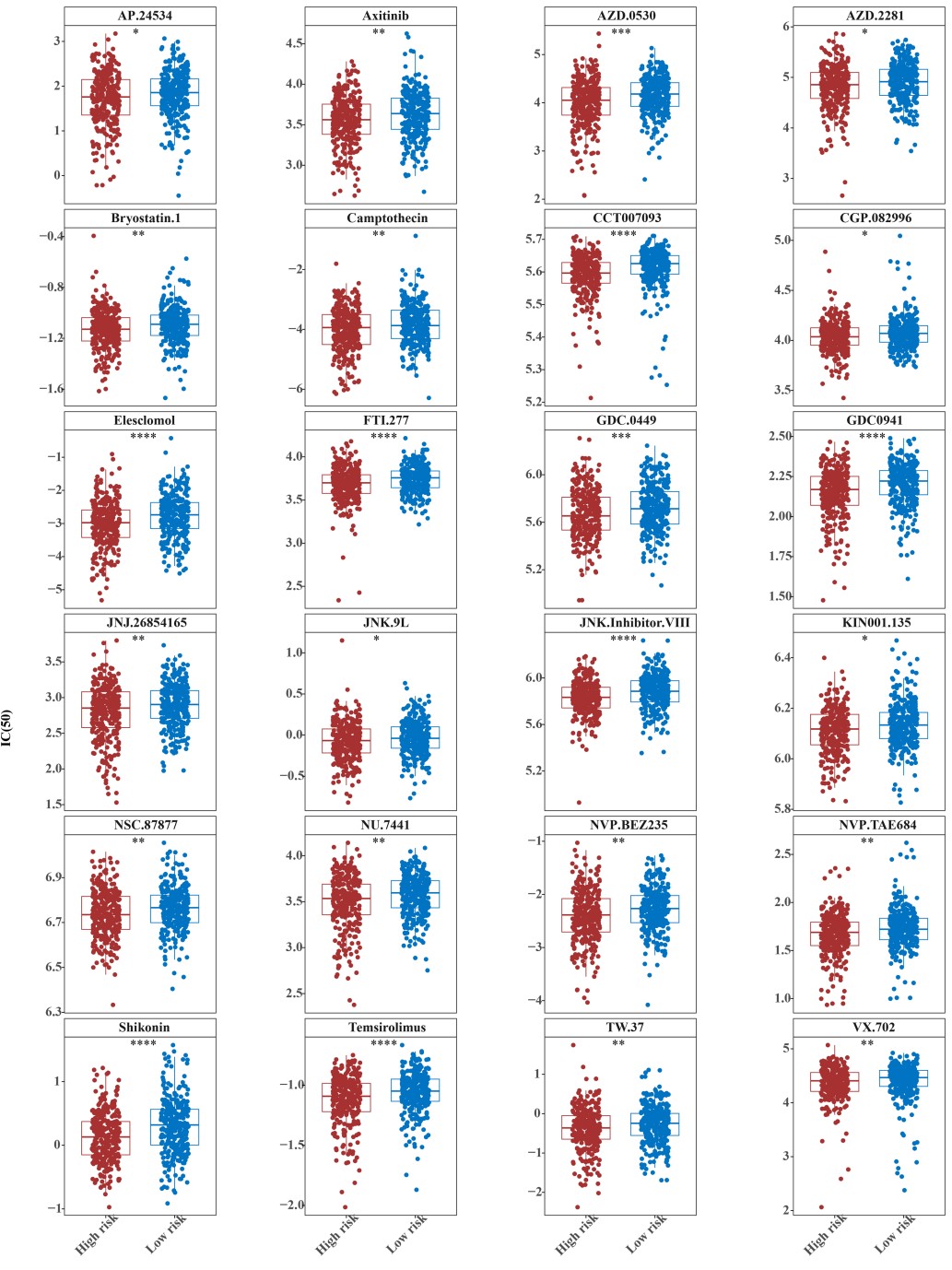

**Figure 9  Twenty-four drugs that perform better in high-risk groups.** The horizontal coordinate is the high-low expression group; the ordinate is IC50.

2019), with caspase activation marking the irreversible point of cell death (*Boatright & Salvesen, 2003*). The combined expression of Caspase-8 and Caspase-3 exhibits synergistic effects and serves as an effective prognostic indicator for patients with CRC (*Jing et al., 2016*). Additionally, p20BAP31 has been shown to induce CRC cell apoptosis *via* the AIF

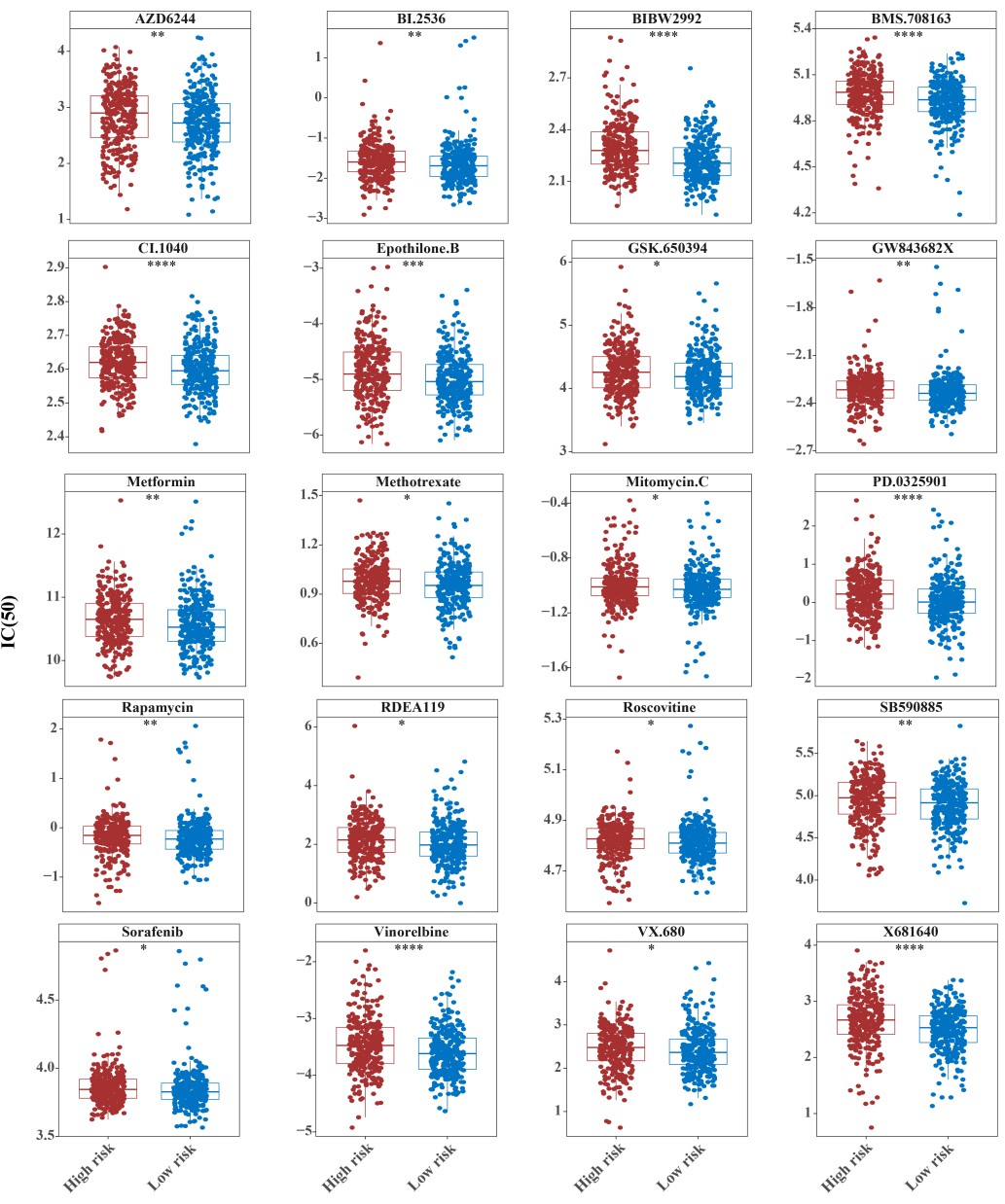

**Figure 10  Ten drugs that perform better in high-risk groups.** The horizontal coordinate is the high and low expression group; the ordinate is IC50.

Caspase-independent and ROS/JNK mitochondrial pathways (*Jiang et al., 2023*). DNA helicases, which catalyze the unwinding of duplex nucleic acids using ATP hydrolysis, are crucial for various DNA-related biological functions (*Hidese et al., 2018*). These helicases not only maintain genome stability but also play significant roles in cancer. Their involvement in DNA damage, replication stress responses, and repair pathways underscores their critical function in cancer biology (*Dhar, Datta & Brosh Jr, 2020*). These

**Table 5  Table of relevant information on 71 discrepant drugs.**

| drugs | high_group_res | low_group_res | padj | pvalue | log2FoldChange | References |
|---|---|---|---|---|---|---|
| Mitomycin.C | −1.011 | −1.029 | 0.09352216 | 0.048116473 | 0.028560435 | PMID: 39531120 |
| Sorafenib | 3.844 | 3.826 | 0.083005791 | 0.042104387 | 0.014156188 | PMID: 39577235 |
| VX.680 | 2.478 | 2.364 | 0.075511125 | 0.037755563 | 0.039769178 | |
| X17.AAG | −0.9516 | −0.8502 | 0.072798168 | 0.035871561 | −0.144174664 | |
| Methotrexate | 0.9782 | 0.953 | 0.064334946 | 0.031235082 | 0.021896508 | PMID: 363377 |
| AZD.2281 | 4.856 | 4.915 | 0.059041856 | 0.028237409 | −0.090080215 | PMID: 36082969 |
| GSK.650394 | 4.256 | 4.187 | 0.055894746 | 0.026327236 | 0.05820392 | |
| CGP.082996 | 4.036 | 4.07 | 0.04924791 | 0.022839611 | −0.033324433 | |
| ABT.263 | 2.787 | 2.855 | 0.042698772 | 0.019492918 | −0.066165992 | |
| MK.2206 | 2.807 | 2.837 | 0.041125831 | 0.018476823 | −0.02663862 | |
| KIN001.135 | 6.118 | 6.134 | 0.033967735 | 0.015014723 | −0.022779748 | |
| JNK.9L | −0.06688 | −0.0392 | 0.031843664 | 0.013845071 | −0.046716942 | |
| AP.24534 | 1.758 | 1.855 | 0.031843664 | 0.01364931 | −0.124897566 | PMID: 38722621 |
| RDEA119 | 2.148 | 1.979 | 0.031843664 | 0.013543565 | 0.117379887 | |
| Roscovitine | 4.827 | 4.81 | 0.025538218 | 0.010548395 | 0.005275309 | |
| TW.37 | −0.3615 | −0.2439 | 0.021059601 | 0.008545925 | −0.112737783 | |
| Camptothecin | −3.94 | −3.873 | 0.021059601 | 0.008499383 | −0.189793129 | PMID: 16990856 |
| Rapamycin | −0.1547 | −0.2284 | 0.021059601 | 0.008429995 | 0.028144422 | PMID: 37057884 |
| JNJ.26854165 | 2.853 | 2.903 | 0.019286462 | 0.007407119 | −0.096412009 | |
| KU.55933 | 5.121 | 5.135 | 0.018646596 | 0.007026254 | −0.018304948 | |
| A.770041 | 3.721 | 3.837 | 0.018080039 | 0.006681753 | −0.125054936 | |
| Axitinib | 3.562 | 3.638 | 0.016782954 | 0.006080781 | −0.070962035 | PMID: 39851927 |
| IPA.3 | 5.828 | 5.935 | 0.016529912 | 0.005869317 | −0.083331087 | |
| AZD8055 | 0.362 | 0.3714 | 0.015447953 | 0.005373201 | −0.013423386 | |
| GW843682X | −2.316 | −2.338 | 0.015309048 | 0.005213951 | 0.008854699 | |
| AZD6482 | 3.629 | 3.789 | 0.015068095 | 0.005022698 | −0.129862683 | |
| NSC.87877 | 6.735 | 6.766 | 0.013382545 | 0.004363873 | −0.024400818 | |
| EHT.1864 | 4.778 | 4.85 | 0.013382545 | 0.004300615 | −0.061736239 | |
| VX.702 | 4.406 | 4.467 | 0.011565574 | 0.003603766 | −0.040968786 | |
| SB590885 | 4.975 | 4.915 | 0.010444852 | 0.003178868 | 0.056102008 | |
| BI.2536 | −1.595 | −1.692 | 0.010444852 | 0.003117351 | 0.07229982 | |
| AZD6244 | 2.899 | 2.721 | 0.010435655 | 0.003024827 | 0.119951365 | |
| Docetaxel | −5.362 | −5.26 | 0.007690955 | 0.002173531 | −0.177536721 | PMID: 39911148 |
| NVP.BEZ235 | −2.387 | −2.266 | 0.007245434 | 0.00199512 | −0.119126921 | |
| NU.7441 | 3.534 | 3.595 | 0.006978244 | 0.001870979 | −0.078773514 | |
| Bryostatin.1 | −1.129 | −1.09 | 0.006422026 | 0.001675311 | −0.037451029 | |
| NVP.TAE684 | 1.689 | 1.721 | 0.006218487 | 0.001577153 | −0.062869173 | |
| ZM.447439 | 2.739 | 2.766 | 0.005392136 | 0.001328497 | −0.029209727 | |
| Metformin | 10.65 | 10.53 | 0.005392136 | 0.001300887 | 0.083124597 | PMID: 39861414 |
| PD.173074 | 4.122 | 4.184 | 0.005392136 | 0.00128624 | −0.080062717 | |
| OSI.906 | 3.718 | 3.84 | 0.005392136 | 0.001217238 | −0.132662293 | |

*(continued on next page)*

**Table 5** (*continued*)

| drugs | high_group_res | low_group_res | padj | pvalue | log2FoldChange | References |
|---|---|---|---|---|---|---|
| Midostaurin | 0.8398 | 0.9802 | 0.004306115 | 0.000936112 | −0.127502408 | PMID: 28644114 |
| Dasatinib | 2.085 | 2.413 | 0.003827739 | 0.00080438 | −0.290838304 | PMID: 36322825 |
| Bexarotene | 4.739 | 4.761 | 0.003827739 | 0.000803033 | −0.029905707 | PMID: 39007945 |
| GDC.0449 | 5.653 | 5.714 | 0.003030732 | 0.000592969 | −0.055388504 | PMID: 24756807 |
| AZD7762 | 0.1399 | 0.3149 | 0.002536853 | 0.000477958 | −0.17295287 | |
| Imatinib | 4.954 | 4.978 | 0.002422395 | 0.00043884 | −0.028950786 | PMID: 39023605 |
| RO.3306 | 4.803 | 4.885 | 0.001404402 | 0.000244244 | −0.099624065 | |
| Epothilone.B | −4.903 | −5.04 | 0.001241353 | 0.000206892 | 0.148448881 | |
| GSK269962A | 3.635 | 3.755 | 0.001241353 | 0.00020276 | −0.18013549 | |
| AZD.0530 | 4.053 | 4.181 | 0.000977892 | 0.00014881 | −0.147212978 | |
| JNK.Inhibitor.VIII | 5.833 | 5.886 | 6.28E−05 | 6.37E−06 | −0.054132883 | |
| Vinorelbine | −3.475 | −3.62 | 0.000415296 | 6.02E−05 | 0.146138882 | PMID: 39776939 |
| WO2009093972 | 2.971 | 3.128 | 6.18E−05 | 5.82E−06 | −0.167943832 | |
| PD.0325901 | 0.22 | 0.01011 | 0.000411263 | 5.66E−05 | 0.170290237 | |
| Shikonin | 0.1304 | 0.3193 | 8.96E−06 | 5.19E−07 | −0.171613126 | |
| Pazopanib | 4.534 | 4.644 | 5.18E−05 | 4.51E−06 | −0.135988084 | PMID: 36877187 |
| Temsirolimus | −1.091 | −1.049 | 0.000333998 | 4.36E−05 | −0.06551927 | PMID: 22861825 |
| CMK | 3.639 | 3.663 | 0.000333998 | 4.13E−05 | −0.022727652 | |
| BMS.708163 | 4.986 | 4.938 | 4.99E−05 | 3.98E−06 | 0.049136545 | |
| Elesclomol | −2.976 | −2.739 | 4.99E−05 | 3.82E−06 | −0.257894071 | |
| GDC0941 | 2.169 | 2.222 | 4.99E−05 | 3.57E−06 | −0.055258206 | |
| FTI.277 | 3.699 | 3.757 | 0.000298753 | 3.46E−05 | −0.061556158 | |
| BIBW2992 | 2.279 | 2.206 | 3.09E−09 | 2.24E−11 | 0.076788254 | PMID: 39894491 |
| X681640 | 2.668 | 2.531 | 4.28E−06 | 2.17E−07 | 0.16932215 | |
| CI.1040 | 2.62 | 2.595 | 0.000163328 | 1.78E−05 | 0.022405468 | |
| GNF.2 | 4.505 | 4.575 | 3.91E−06 | 1.70E−07 | −0.083372586 | |
| AMG.706 | 4.112 | 4.193 | 7.28E−09 | 1.58E−10 | −0.104445814 | |
| AG.014699 | 4.157 | 4.187 | 4.37E−08 | 1.27E−09 | −0.040279378 | PMID: 30830551 |
| LFM.A13 | 6.142 | 6.205 | 2.83E−07 | 1.02E−08 | −0.065518629 | |
| CCT007093 | 5.597 | 5.626 | 7.03E−09 | 1.02E−10 | −0.024293532 | |

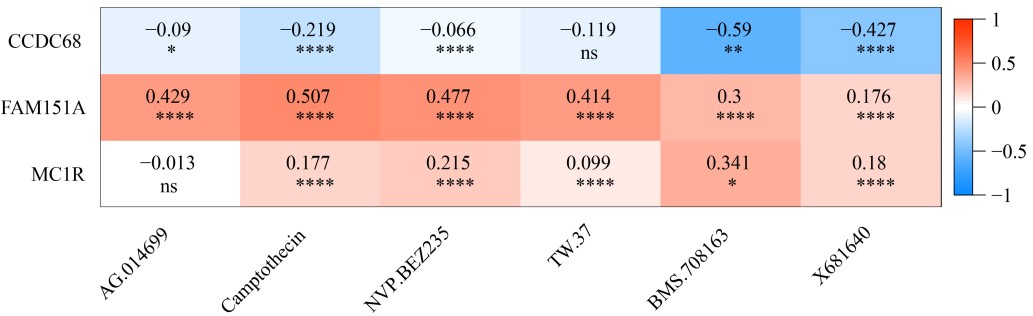

**Figure 11** Gene correlation between chemotherapy drugs and risk model.

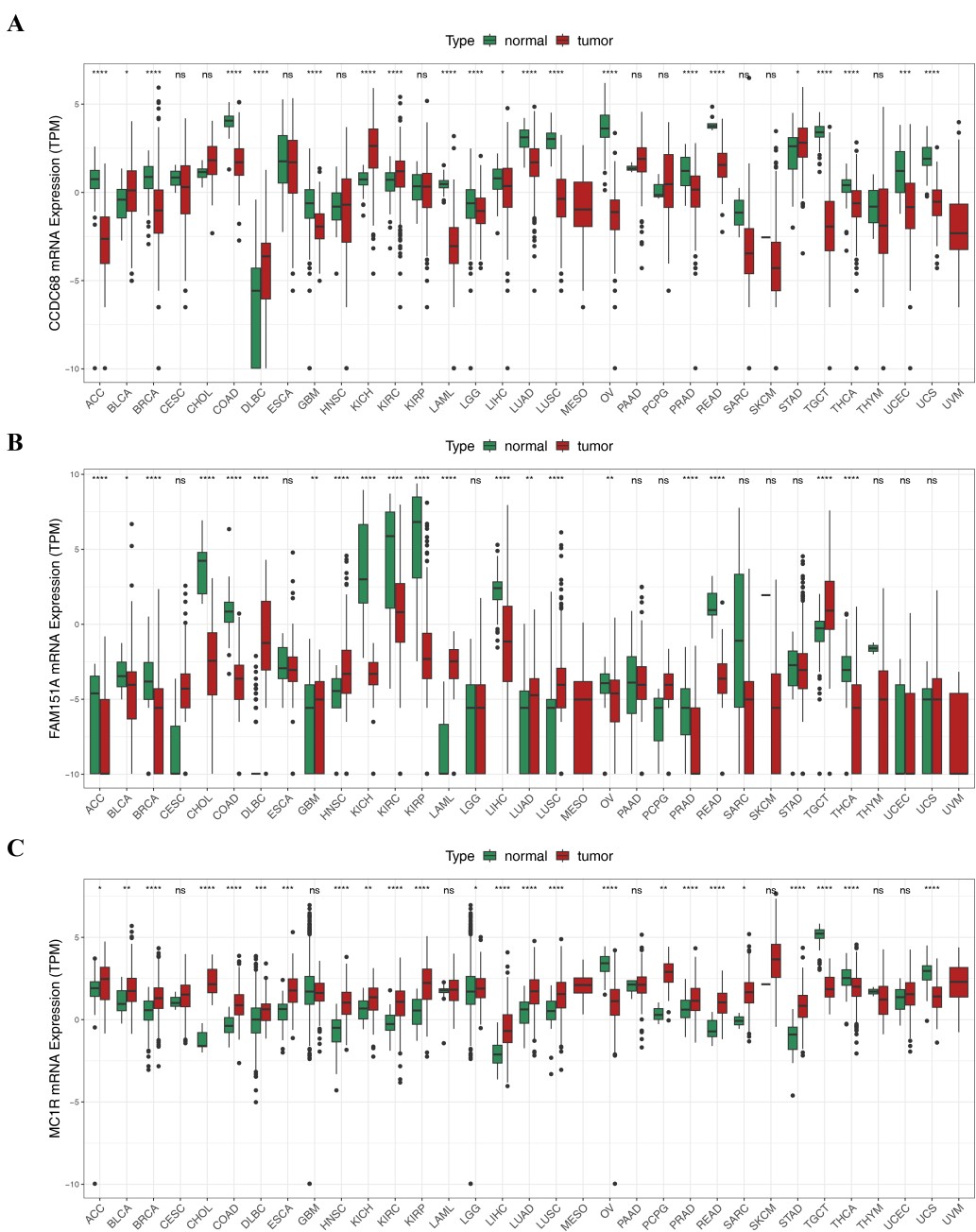

**Figure 12 Three hub genes linked in other diseases.** (A) Differential analysis of CCDC68 in different samples of pancarcinoma. (B) Differential analysis of FAM151A in different samples of pancarcinoma. (C) Differential analysis of MC1R in different samples of pancarcinoma.

findings provide valuable insights into the mechanistic understanding of CRC onset and offer potential avenues for its prevention and treatment.

Three key nuclear genes—*CCDC68*, *FAM151A*, and *MC1R*—were identified through the construction of a risk model, with their prognostic significance assessed and validated. *CCDC68* may contribute to CRC pathogenesis *via* multiple signaling pathways, and

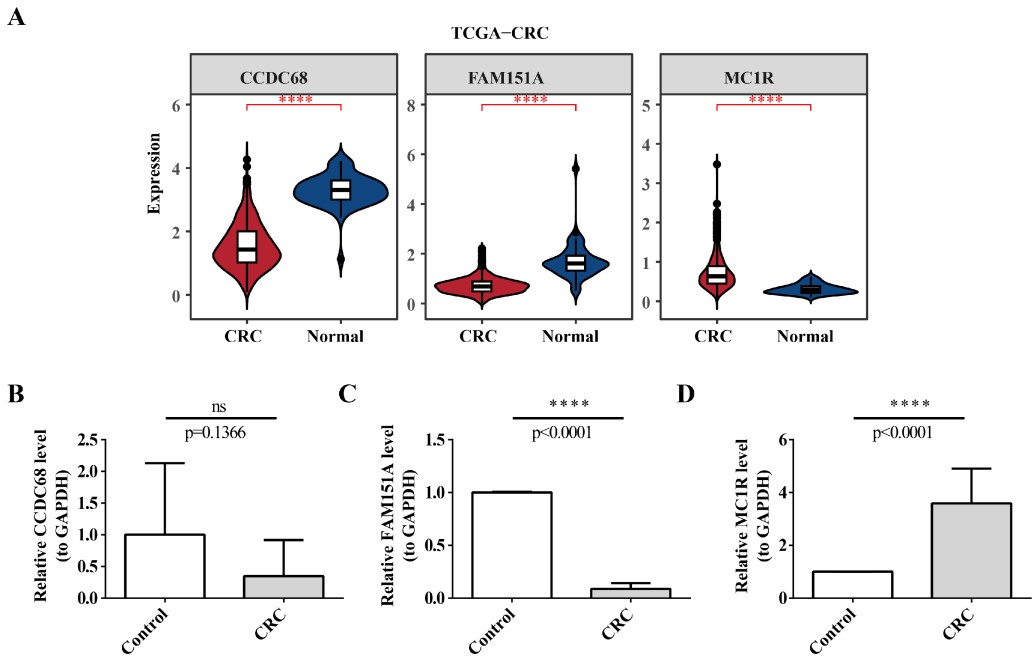

**Figure 13** **Validation of expression of 3 biomarkers.** (A) Difference analysis of risk model genes in different samples of training set. (B) Expression of CCDC68 in normal samples and CRC samples. (C) Expression of FAM151A in normal samples and CRC samples. (D) Expression of MC1R in normal samples and CRC samples.

functional experiments demonstrate its inhibitory effects on CRC cell growth *in vitro* and tumor formation *in vivo* (*Wang et al., 2021*). While *CCDC68* acts as a pro-carcinogenic factor in IL-6-stimulated endometrial cancer cells (*Li et al., 2021*), it also exhibits tumor-suppressor properties in pancreatic ductal adenocarcinoma (*Radulovich et al., 2015*). Although *CCDC68* has not been directly implicated in mitochondrial division, its potential role in cytoskeleton, signaling, and gene expression regulation may indirectly affect mitochondrial morphology and function (*Huang et al., 2017*). *FAM151*, a member of the PLC-like phosphodiesterase superfamily, remains enigmatic, with its substrate and function yet to be identified (*Findlay et al., 2020*). No research has been conducted on the impact of *FAM151A* in CRC; however, *Fam151B* homozygous knockout mice develop retinal degeneration, with signs of retinal stress and rapid loss of photoreceptor cells in the eye, while *FAM151A* homozygous mutant mice have no discernable phenotype, suggesting that *Fam151b* and *FAM151A* may be functionally different (*Findlay et al., 2020*). Although no direct evidence links *FAM151A* to mitochondrial fission, it may influence mitochondrial dynamics through signal transduction, cellular structural interactions, and gene expression regulation, warranting further investigation. In addition, *FAM210B* participates in the mitochondrial energy metabolism of erythroblasts and makes a prominent contribution to erythrocyte differentiation by regulating mitochondrial energy metabolism (*Suzuki et al., 2022*). *MC1R*, a member of the G protein-coupled receptor (GPCR) subfamily, regulates key physiological and behavioral features *via* melanocortin binding, making

it a potential target for melanoma therapy (*Guida, Guida & Goding, 2022*). High *MC1R* expression is significantly associated with microsatellite instability (MSI) (*Peng et al., 2021*). *MC1R* signaling accelerates G1/S phase progression and promotes breast cancer progression through the cAMP-CREB/ATF-1 and ERK-NFκB pathways (*Chelakkot et al., 2023*). Polymorphisms in the *MC1R* gene, leading to red pigmentation, are linked to a reduced risk of prostate cancer (*Weinstein, Virtamo & Albanes, 2013*), and *MC1R* is highly expressed in esophageal squamous cell carcinoma (*Zhou et al., 2022*). Furthermore, *MC1R* plays a pivotal role in CRC progression and may serve as a marker of poor prognosis in CRC (*Peng et al., 2021*). Existing research suggests that *MC1R* may influence mitochondrial fission through two primary mechanisms. The first involves oxidative stress regulation, as *MC1R* plays a role in modulating oxidative stress responses in melanocytes (*Lu et al., 2024*). Mitochondria serve as the primary source of intracellular reactive oxygen species (ROS), and the balance between mitochondrial fission and fusion is essential for maintaining mitochondrial function and cellular redox homeostasis (*Westermann, 2012*). *MC1R* activity may influence mitochondrial function, thereby indirectly impacting mitochondrial fission. The second mechanism is mediated through cellular signaling pathways. *MC1R* activation by α-MSH directly triggers cAMP signaling, leading to AMPK activation (*Sun et al., 2023*). Under energy stress, AMPK localizes to mitochondria, where it phosphorylates Ser637 of mitochondrial dynamic associated protein 1 (DRP1) and Ser155/Ser172 of its receptor, mitochondrial fission factor (MFF), facilitating DRP1 recruitment to the outer mitochondrial membrane and inducing fission (*Hsu et al., 2022*). Additionally, *MC1R* deficiency has been linked to metabolic dysregulation. *MC1R*-knockout mice exhibit significant hepatomegaly, accompanied by elevated hepatic and plasma cholesterol and triglyceride levels, suggesting that hepatocyte *MC1R* signaling regulates cholesterol and bile acid metabolism, while its absence promotes hypercholesterolemia (*Thapa et al., 2023*). Furthermore, some downregulated genes following *MC1R* knockout have been implicated in cell migration and melanoma metastasis (*Seong & Kim, 2014*), but its role in CRC cell migration and metastasis remains uncharacterized and warrants further investigation. This study reveals a strong association between *CCDC68*, *FAM151A*, and *MC1R* in CRC development, suggesting a potential therapeutic strategy. The study model demonstrates a correlation with N stage and M stage in CRC, and *CCDC68* expression emerges as a promising molecular marker for prognostic evaluation in patients with CRC.

Cancer development and progression are closely linked to alterations in the tumor microenvironment. Cancer cells actively reshape their surroundings by secreting various cytokines, chemokines, and other factors, which reprogram the neighboring cells to support tumor survival and progression (*De Visser & Joyce, 2023*). Immune cells, key components of the tumor stroma, play a critical role in this dynamic process (*Hinshaw & Shevde, 2019*). This study assessed the relative proportions of 22 immune cell types in high- and low-risk CRC subgroups, evaluating both immune cell infiltration and immune function scores. To investigate differences in immune infiltration between these subgroups, Wilcoxon tests were performed. The results revealed five immune cell types with significant differences between the high- and low-risk groups: resting dendritic cells, resting memory CD4+ T cells, activated memory CD4+ T cells, plasma cells, and M0

macrophages. Dendritic cells are pivotal in regulating adaptive immune responses and are essential for T cell-mediated cancer immunity (*Gardner & Ruffell, 2016*). They play a central role in initiating and modulating both innate and adaptive immune responses (*Wculek et al., 2020*). Memory CD4+ T cells are involved in the immune response and have been associated with better prognosis in various cancers (*Liu et al., 2021*). Plasma cells, key effectors of adaptive immunity, produce antibodies that protect the body from pathogens (*Varlet et al., 2020*). Evidence suggests that plasma cells actively contribute to anti-tumor immunity (*Wouters & Nelson, 2018*). M0 macrophages are predominantly infiltrated in high-risk patients with CRC, where they suppress tumor immune responses and correlate with poor survival (*Castrogiovanni et al., 2022*; *Dong et al., 2024*). *MC1R* has been shown to stimulate HLA-A2-restricted cytotoxic T lymphocytes, enhancing their ability to recognize peptides naturally processed on melanoma cells, further linking immune modulation to tumor immunity (*Salazar-Onfray et al., 1997*).

This study investigated the relationship between chemotherapy drug sensitivity and gene expression in a risk model. Notably, *FAM151A* exhibited a positive correlation with four drugs (AG.014699, camptothecin, NVP.BEZ235, TW.37) ($r > 0.4$), while *CCDC68* was positively associated with two drugs (BMS.708163, X681640) ($r < -0.4$). AG.014699 (rucaparib), a poly(ADP-ribose) polymerase inhibitor, was shown to protect Schwann cells from cell death and reduce glycolysis, though it did not counteract the disruption of the TCA cycle under high-glucose conditions in the absence of pyruvate (*Yako et al., 2024*). Relevant studies (*Bugajova et al., 2024*) suggest that NVP.BEZ235 can reduce ATP production. Treatment with NVP.BEZ235 increases the mitochondrial autophagy-related protein BNIP3, promoting mitochondrial fission, enhancing protease activity, and facilitating the degradation of mitochondrial proteins. Camptothecin, a potent topoisomerase inhibitor, was found to increase the proportion of cells with decreased mitochondrial membrane potential, correlating with lower ATP levels. In colorectal carcinoma cell lines (DLD1 and HCT-116), camptothecin elevated apoptosis rates and significantly reduced cell viability after 24 h of treatment by lowering ATP production and pyruvate levels (*Liskova et al., 2022*). In conclusion, NVP.BEZ235 may influence mitochondrial fission through BNIP3-mediated mitochondrial autophagy, while camptothecin induces mitochondrial dysfunction and fission by reducing mitochondrial membrane potential and ATP levels. The main mechanism of action of AG.014699 (rucaparib) is not directly related to mitochondrial fission, but its regulation of cell metabolism may indirectly affect mitochondrial function.

Although this study has made some progress, there are still some limitations. First, the predictive power of the model (AUC value of about 0.6) has room for further improvement and needs to be optimized and improved by introducing more and more predictive features. Secondly, there is a lack of further *in vivo* and *in vitro* experiments for functional verification, especially the exploration of the specific role and mechanism of MC1R gene in colorectal cancer (CRC) is still insufficient. Therefore, future studies should validate the model predictions through clinical data and *in vivo*/*in vitro* experiments (such as mouse models, cell models, gene knockout or overexpression, drug treatment, *etc.*), and further explore the mechanism by which MC1R regulates mitochondrial fission through cAMP/AMPK/DRP1/MFF axis and promotes CRC development. In addition, further

exploration of the functional role of these key genes in CRC progression, as well as their potential value in therapy, is of great significance for the discovery of new immunotherapy and targeted therapy strategies.

## CONCLUSIONS

In this study, a series of bioinformatics methods were used to identify the prognostic value of mitochondrial fission-related genes in CRC. Three genes (*CCDC68*, *FAM151A*, *MC1R*) were identified as potential risk model genes in CRC. The risk model was constructed, and the molecular mechanism of mitochondrial fission-related genes affecting CRC was further analyzed, providing a new direction for CRC treatment.

## ACKNOWLEDGEMENTS

We would like to express our sincere gratitude to all individuals and organizations who supported and assisted us throughout this research. Without your support, this research would not have been possible.

### Funding

This work was supported by the Autonomous Region Health Commission Self-funded Research Project (Z-A20240028). The funders had no role in study design, data collection and analysis, decision to publish, or preparation of the manuscript.

### Grant Disclosures

The following grant information was disclosed by the authors:
The Autonomous Region Health Commission Self-funded Research Project: Z-A20240028.

### Competing Interests

The authors declare there are no competing interests.

### Author Contributions

- Chao Liu conceived and designed the experiments, performed the experiments, analyzed the data, authored or reviewed drafts of the article, and approved the final draft.
- Sheng Xu performed the experiments, analyzed the data, prepared figures and/or tables, and approved the final draft.
- Yuanyuan Liu performed the experiments, prepared figures and/or tables, and approved the final draft.
- Zhixing Lu performed the experiments, prepared figures and/or tables, and approved the final draft.
- Jianrong Yang conceived and designed the experiments, authored or reviewed drafts of the article, and approved the final draft.

## Human Ethics

The following information was supplied relating to ethical approvals (i.e., approving body and any reference numbers):

The People's Hospital of Guangxi Zhuang Autonomous Region granted Ethical approval to carry out the study within its facilities (Ethical Application Ref: KY-ZC-2024-030).

## Field Study Permissions

The following information was supplied relating to field study approvals (i.e., approving body and any reference numbers):

Field experiments were approved by the Research Council of the People's Hospital of Guangxi Zhuang Autonomous Region (project number: KY-ZC-2024-030).

## Data Availability

The raw measurements are available in the Supplemental Files.

The datasets analyzed are available at GEO, GSE103479; TCGA (search terms TCGA, TCGA-COAD and TCGA-READ, RNA-Seq, transcriptome profiling); MSigDB, (search terms: Mitochondrial Fission; GOBP_MITOCHONDRIAL_FISSION and GOBP_POSITIVE_REGULATION_ OF_MITOCHONDRIAL_FISSION).

## Supplemental Information

Supplemental information for this article can be found online at http://dx.doi.org/10.7717/peerj.19522#supplemental-information.

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
