# Peer review of "Mito-fission gene prognostic model for colorectal cancer"

_PeerJ, doi:10.7717/peerj.19522_

## Round 0.1 · original submission · Major Revisions

Thank you for your submission to PeerJ. Please address these changes in a clear rebuttal and revision.

Reviewer 1 ·

Basic reporting

The manuscript demonstrates clear and professional use of English, with a focused introduction linking mitochondrial fission to colorectal cancer (CRC). The context and knowledge gaps are well-established, supported by relevant and recent references. Figures are high quality, appropriately labeled, and effectively convey the study's findings, such as gene expression and risk model validations. Raw data availability ensures transparency and adherence to journal standards. Overall, the reporting is strong, with minor potential improvements in grammar, style, and additional references for enhanced robustness.

Experimental design

The study uses public databases (TCGA, GEO) and bioinformatics methods to analyze mitochondrial fission-related genes (MFRGs) in colorectal cancer (CRC). Differentially expressed genes (DEGs) and weighted gene co-expression network analysis (WGCNA) were applied to identify hub genes. A prognostic risk model was constructed using statistical techniques like LASSO and Cox regression and validated with an independent dataset (GSE103479). Clinical correlations, gene enrichment, immune profiling, and drug sensitivity analyses were included to explore biological relevance and therapeutic implications. The methods are rigorous, reproducible, and ethically approved, aligning well with study goals.

Validity of the findings

The findings are supported by robust bioinformatics analyses, validated through an independent dataset (GSE103479), ensuring reproducibility. The identified hub genes (CCDC68, FAM151A, MC1R) and the constructed risk model demonstrate statistical significance in predicting CRC prognosis, with clear correlations to clinical features. Enrichment and immune profiling analyses further validate the biological relevance. However, additional in vivo and in vitro experiments are needed to fully substantiate the functional roles of these genes in CRC progression and therapy. The study's conclusions are consistent with the presented data, providing a solid theoretical foundation for future research.

Additional comments

nil

Reviewer 2 ·

Basic reporting

The authors attempted to the role of mitochondrial fission genes in colorectal cancer (CRC). By analyzing data from the TCGA-CRC and GSE103479 datasets and using bioinformatics approaches, the authors reported three hub genes (CCDC68, FAM151A, MC1R) associated with CRC progression. A risk model based on these genes demonstrated significant prognostic value, with higher risk scores correlating with worse survival outcomes. The study also explored immune infiltration patterns and drug sensitivity, proposing these findings as potential directions for CRC treatment. In general, the study may provide some insight into the role played by mitochondrial fission genes in CRC, but there were many drawbacks in it. Specific comments are given as follows:

1. The results and the figures were poorly organized and hard to follow.
2. There were many grammar errors in the manuscript. Please check it carefully
3. The full names of genes were not mentioned.
4. The format of gene symbols was not consistent.
5. "P" for p-values should be italicized.

Experimental design

1. The most important concern was that the DE-mitochondrial fusion-related genes (MFRGs) identified in the study were, in fact, DEGs correlated with the MFRG score. It was unclear why and how they could play a role in mitochondrial fusion. Why not focus on the 40 MFRGs that were differentially expressed in CRC? What were these 40 MFRGs identified?
2. What were the criteria for selecting the three hub genes (CCDC68, FAM151A, MC1R)?
3. What can be concluded based on the drugs showing sensitivity in the patients of different risk groups? Did these drugs possibly target the mitochondrial fusion process?

Validity of the findings

The prognostic performance of the risk model was not good enough, with AUCs around 0.6 only.

·

Basic reporting

No Comment

Experimental design

1) The authors should provide more details about the statistical analysis used in this study.

Validity of the findings

I read the "Mito-Fission Gene Prognostic Model for Colorectal Cancer" research article with great interest.
In this study, the authors used a set of bioinformatics approaches to determine the prognostic value of mitochondrial fission-related genes in colorectal cancer patients. They used the Cancer Genome Atlas (TCGA)-CRC dataset as well as the GSE103479 dataset (validation of risk model) covering 40 mitochondrial Fission-related genes (MFRGs). The data highlighted the role of three genes (CCDC68, FAM151A, MC1R) as potential risk model factors in CRC. They further look at the constructed risk model and the molecular mechanism of mitochondrial fission-related genes impacting the progression of CRC. This study provides a new platform for understanding the role of mitochondria in CRC formation, progression, and treatment.

Additional comments

Below is a list of comments and concerns that I hope the authors will find helpful in improving this manuscript.

1) Figure 1E showed no significance in their MFRG score among CRC patients below and above 60 years old. What was the rationale behind the 60-year cut-off since "early-onset colorectal cancer (CRC)" refers to a diagnosis of colorectal cancer in patients under the age of 50?
2) Authors conducted a univariate Cox analysis, which showed 10 genes related to survival (P-value < 0.05), of which HIGD1A, DIAPH3, CCDC68, and FAM151A are protective factors, but KPTAP5.1, HSF4, MC1R, ZNF692, LIME1, and TTYH3 are risk factors (Fig. 4A). What are the relations/functions/roles of these protective and risk factors in terms of mitochondrial dysregulation in CRC or malignant tumors in general. A table with a brief explanation of their function in cancer cells and their corresponding references elevate the quality of this finding.
3) Figure 7 indicates the effect of three hub genes on the immune microenvironment. Figure 7E demonstrates that the TIDE score and immune exclusion score were more meaningful in evaluating the effectiveness of immune therapy. The authors should clarify the outcomes of the results illustrated in Figure 7 with more details.
4) The authors examined several common chemotherapeutic drugs and their effectiveness in high-risk patients for CRC. Figures 8 to 10 revealed that the IC50 of 71 chemotherapeutic drugs significantly differed between the two risk group samples. Additionally, only 51 drugs showed better response in the high-risk group patients. How many of drugs in these 51 drugs are currently prescribed for patients with CRC? A separate table with a list of these drugs elevates the quality of this set of experiments.
5) There is evidence for the contribution of FAM151A and MC1R to mitochondrial metabolism and biogenesis, such as PMID: 36374104. Does the CCDC68 protein have a defined impact on mitochondrial function or its dysregulation?
6) Discussion section needs grammatical issues such as line 333 (… that occurs in thepost-mitosis period …), line 336 (… specific son of cytoplasmic division…), line 368 (…stability but also in cancer.. The …) and line 388 (… (Chelakkot et al., 2023). Red publication caused by polymorphisms in the MC1R …).
7) The authors concluded, "However, the results still need to be further verified by in vivo and in vitro experiments. We will conduct further experiments to explore these key genes' roles and mechanisms.” Are there any previous cell or animal models where the loss of function of these three individual genes (genes has been described phenotypically?

---

## Round 0.2 · Minor Revisions

Thanks for your work to PeerJ.

Please change your manuscript as the comments from reviewers.

·

Basic reporting

No further comment

Experimental design

No further comment

Validity of the findings

No further comment

Additional comments

Authors satisfactorily addressed all raised comments and questions.

Reviewer 4 ·

Basic reporting

The authors aimed to investigate the role of mitochondrial fission genes in CRC. By analyzing data from the TCGA-CRC and GSE103479 datasets and employing bioinformatics methods, the authors identified three key genes (CCDC68, FAM151A, and MC1R) associated with CRC progression. The risk model based on these genes demonstrated significant prognostic value, with higher risk scores correlating with poorer survival outcomes. This study may provide insights into the role of mitochondrial fission genes in CRC.

Experimental design

The methodology should provide a detailed description of the MTFRG score calculation to enhance the interpretation of Result 2. This is particularly important as the methodology section specifies that there are 40 MFRG-related genes (as shown in Table 1), whereas Result 2 identifies 49 DE-MFRGs.

Validity of the findings

1)The main focus of the article is to investigate the impact of mitochondrial fission-related genes on CRC. Providing annotations for the associated genes (e.g., their origin, whether mitochondrial or nuclear) would aid readers in better understanding the findings.
2)In Result 3.4, the advantages of the three hub genes (CCDC68, FAM151A, and MC1R) selected for the construction of the prognostic risk model should be articulated more clearly, with a more comprehensive and precise description of the results.
3)The description in Result 3.5 is unclear. What is the basis for categorizing the risk score into high and low-risk groups?
4)In Result 3.7, what is the association between the high- and low-risk groups and MFRGs?

---

## Round 0.3 · accepted · Accept

Thank you for your submission to PeerJ.
Congratulations!

Reviewer 4 ·

Basic reporting

No comments

Experimental design

No comments

Validity of the findings

No comments

Additional comments

No comments